# An In-Depth Analysis of Physical Blue and Green Water Scarcity in Agriculture in Terms of Causes and Events and Perceived Amenability to Economic Interpretation

**Kalomoira Zisopoulou [1] and Dionysia Panagoulia [2,***

1   Travaux Publics, Becket House, London SE1 7EU, UK; kal.evg.zisopoulou@gmail.com
2   Department of Water Resources and Environmental Engineering, School of Civil Engineering, National Technical University of Athens, Zografou, 157 80 Athens, Greece
*   Correspondence: dpanag@hydro.ntua.gr

**Abstract:** An analytical review of physical blue and green water scarcity in terms of agricultural use, and its amenability to economic interpretation, is presented, employing more than 600 references. The main definitions and classifications involved and information about reserves and resources are critically analyzed, blue and green water scarcity are examined along with their interchange, while their causal connection with climate in general is analyzed along with the particular instances of Europe, Africa, Asia and the WANA region. The role of teleconnections and evaporation/moisture import-export is examined as forms of action at a distance. The human intervention scarcity driver is examined extensively in terms of land use land cover change (LULCC), as well as population increase. The discussion deals with following critical problems: green and blue water availability, inadequate accessibility, blue water loss, unevenly distributed precipitation, climate uncertainty and country level over global level precedence. The conclusion singles out, among others, problems emerging from the inter-relationship of physical variables and the difficulty to translate them into economic instrumental variables, as well as the lack of imbedding uncertainty in the underlying physical theory due to the fact that country level measurements are not methodically assumed to be the basic building block of regional and global water scarcity.

**Keywords:** blue water; green water; scarcity; climate; water availability; inadequate water accessibility; climate uncertainty; land use land cover change; population

## 1. Introduction

Water, an economic good [1,2] and "total social fact" [3], is a critical resource [4] as it is a component of human life and ecosystem support and it lies at the base of Maslow's pyramid of human needs [5], while quantitatively/qualitatively is one of the three components of water security [6]. The water scarcity case, expressed in terms of water supply crises, is the number one global societal risk in terms of impact, even greater that the spreading of infectious diseases, and is expected to grow by 43% in 2025 according to the 2015 World Economic Forum [7]. In addition, it will impact >56.2% of the global population by 2080 [8] in terms of societal risks, with both high impact and high likelihood, as seen in Figure 1.

Water scarcity is considered to be the result of a complex interaction of social, economic, and environmental factors, and is seldom caused solely by a lack of precipitation [9]; people may be moved to water resources [10,11] (case studies in [12]) or water resources may be moved to people as, e.g., in the economic development of the Western USA [13] or by reinforcing green water by allowing water surplus to infiltrate into the root zone [14]. Added to these, water consumption is dependent on population growth [14] and climate change [15,16], which intensify scarcity, shocks, and access inequalities and induce physical, financial, regulatory, and reputational risks to businesses [9]. At the same time, water distribution does not match population concentrations, groundwater supplies are dwindling,

and the water multiplier effect [17] does not work in underdeveloped countries, which have no extensive recycling facilities, if at all.

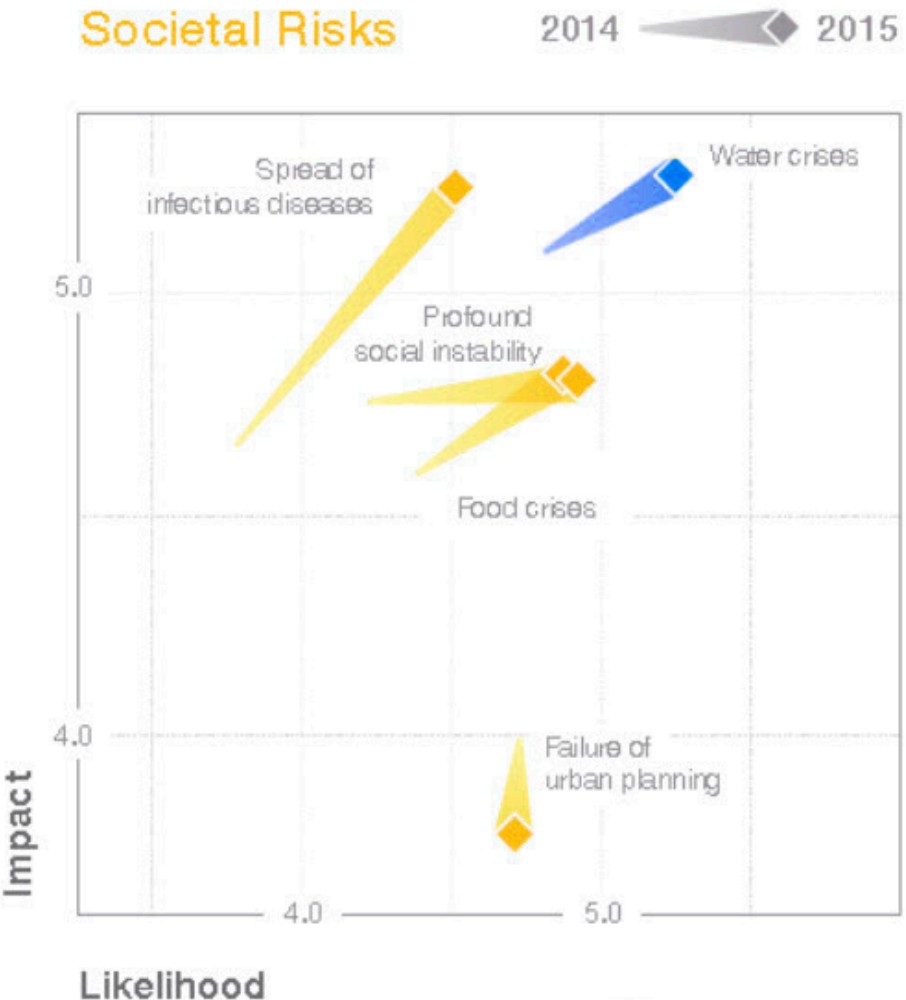

**Figure 1.** Impact-Likelihood diagram of water crises (modified from [7]).

Alternatively, the demand and supply [18] aspect of overpopulation [19], agriculture [20], pollution of water [21], and improper government policies [22], the last of which potentially leading to a case of the "Green Paradox" [23], are important reasons for water scarcity, while land use and land cover change, encompassing crops, livestock, fisheries, aquaculture and forestry, are causes and a victims of water scarcity [16,24,25]. The Comprehensive Assessment of Water Management in Agriculture which is influenced by demand and supply in [18], concludes that water scarcity is a major global constraint to agriculture [26] as well as a growing risk to business and investors [27].

There are two types of water scarcity: physical scarcity and economic scarcity [18]. Physical scarcity is said to occur when water cannot satisfy all demands (including environmental flows) [28], either despite or because of the fact that the global system is interconnected hydro-climatically [29]. Economic scarcity is described as a situation where the socio-economic system is unable to utilize existing water in order to satisfy all demands [30], lacking, in essence, the infrastructure development that consists of storage and timely distribution and access [31], alternatively if human, institutional and financial capital limit access to water [32,33]. Physical and economic scarcity are shown in Figure 2.

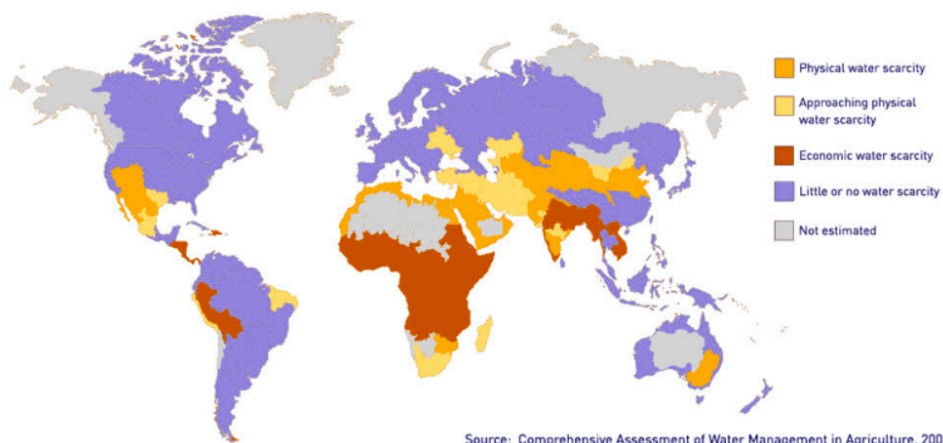

**Figure 2.** Global map of regions with Physical or Economic scarcity (modified from [34]).

Physical water scarcity, the major global management problem of the 21st century [35], causes, among other problems, environmental degradation [36], a decline in groundwater (including subsurface water occurring beneath the water table in soils and geologic formations that are fully saturated [37,38]), and inequitable water distribution [39]. Imposing blue water limitations for agriculture and relying increasingly on rain fed green water has consequences, e.g., these limitations, if imposed on the Western U.S., China and West, South, and Central Asia, would shift 20–60 Mha of cropland to rainfed agriculture, at a loss of 600–2900 Pcal in food production, by the end of this century [40].

Socio-economic drought definitions associate the supply and demand of some economic goods with elements of meteorological, hydrological, and agricultural drought, which occurs when the demand for an economic good exceeds supply as a result of a weather-related shortfall in the water supply [41]. Water scarcity is measured by a variety of indicators/metrics, reviewed in [42,43], which are quantified in [26] and criticized in [44]. Of these, the indicator of the blue water sustainability index (BlWSI) [45] is of particular interest as, beyond consumptive blue water use (CBWU), it includes non-renewable groundwater abstraction (NRGW$_A$) and non-satisfied environmental streamflow.

The purpose of this analytical review is to establish a platform of physical blue and green water scarcity characteristics with respect to agriculture, classified according to their causes and impact by employing concrete examples indicative of the characteristic's spatio-temporal spectrum, which are useful in analyzing the economics of water scarcity for blue and green water, focusing on agriculture. Section 3.1 describes the variety of existing pertinent definitions and classifications; Section 3.2 analyzes blue and green water scarcity; Section 3.2. examines blue and green water interactions with climate, including short analyses of Europe, Africa, Asia, and the WANA region.; Section 3.4 addresses the subject of green and blue water impact by actions at a distance, teleconnections and evaporation and moisture import/export. Section 3.5 Discusses important points regarding the problems encountered.

## 2. Methodology

The objective of this paper is to present blue and green water scarcity, both in general case and examples in terms such that a manifest relation can be established between definitions and aspects of the physical causes and impacts of scarcity with respect to agriculture on the one hand, and instrumental economic variables that may lead to economic results and models on the other. In Figure 3, a simplified process of a country level physical event is presented, where the main sequence is global climate to local climate to scarcity event, and in the end to water stakeholders. As intermediaries between global and local climate are climate teleconnections, evaporation moisture import–export, which is in bidirectional feedback mode, and direct intervention; between the scarcity event and water stakeholders are physical and economic scarcity processes. One of the contributors, human intervention,



is broken into two parts, global and local, connected by bidirectional feedback. The curved arrows show the connections this paper aims to facilitate.

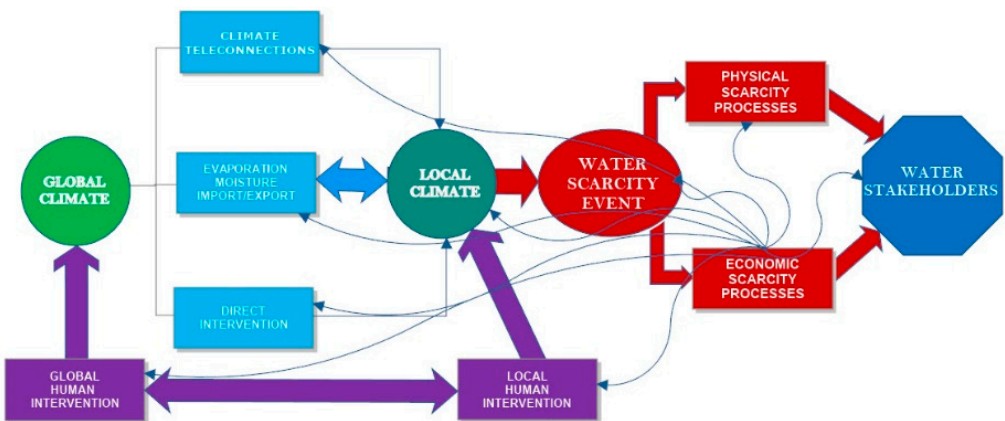

**Figure 3.** A simplified process of country level water scarcity event.

Despite the simplicity of the diagram, it is easy to see that the causes and effects of physical water scarcity form in effect a multifaceted construct, where localized country physical conditions may lead to either similar or different results for the same physical phenomenon, hence the large number of references.

The stages shown below expose as many of the multiple aspects of this scarcity as possible and their causal interconnections with widely acceptable mainline physical results using concrete events described in the relevant literature. The use of these events also serves the purpose of opening an avenue that serves researcher who wishes to assess and evaluate, in economic terms, the class of events to which a particular event belongs.

The first column in Figure 4 refers to the general case of physical phenomena judged to be pertinent on the basis of normative decision and its row expansions refer to particular influential phenomena while their content includes special cases.

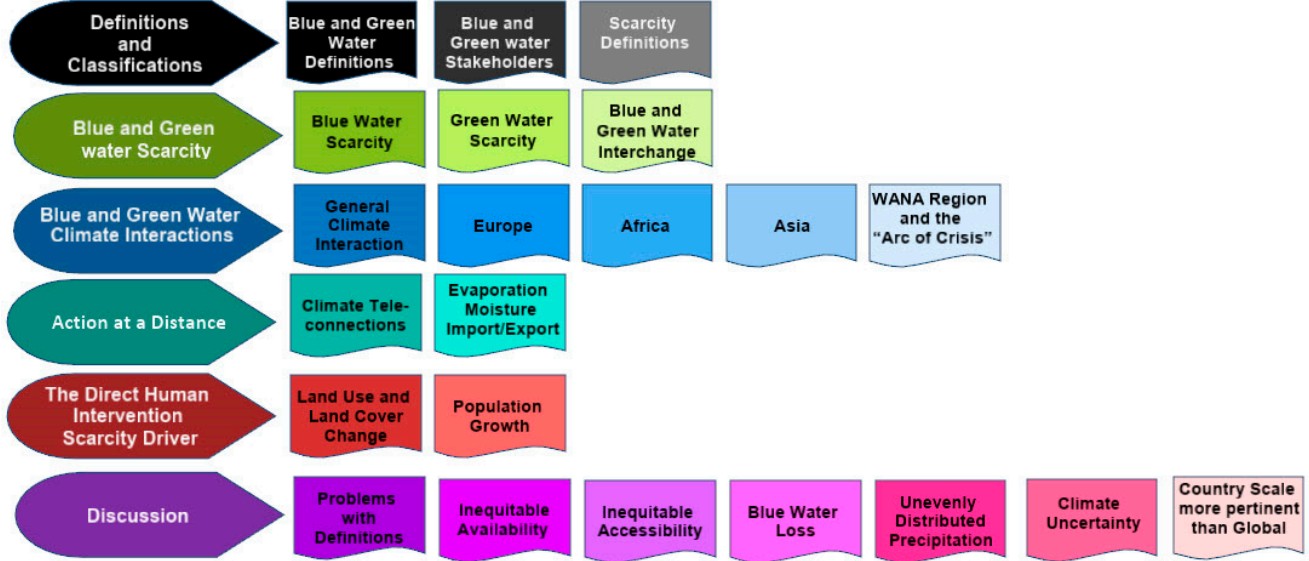

**Figure 4.** The stages of Methodology.

## 3. Results

### 3.1. Definitions and Classifications

#### 3.1.1. Blue and Green Water Definitions

A variety of definitions for both blue and green water exist [46]. Falkemark's original definition was that incoming rainfall is partitioned into vertical upward flow and horizontal flow, leading to aquifers and rivers, which constitutes "blue water" while "green water" is the water in the root zone, the part of the upper vadose zone instrumental in partitioning rain and irrigation water into evaporation, transpiration, runoff, and deep drainage [47], which is the source of plant nutrition [48]. A later, more precise definition [49], was given in terms of resource supply: blue water is the water in aquifers, lakes, and dams, and green water is the moisture in the soil which are related to the liquid blue water flowing through rivers and aquifers and the green water vapor flowing back to the atmosphere. Green water is divided into two parts [49–51], one part is stored in the soil as moisture and another part is in motion via the evapotranspiration process.

Blue water can be classified according to its state: in liquid flow form, stocked as runoff, rivers, reservoirs, wetlands, lakes, snowpack, aquifers for the consumptive pathway of household or industrial uses, drinking water and product integration, and in vapour flow form, stocked as surface water or groundwater for the consumptive pathway of evapotranspiration from irrigation [49,52]. Green water can be classified in the same way: in vapour flow, as productive green water is stocked as soil moisture for the consumptive pathway of plant transpiration, and in vapour flow as unproductive green water stocked as soil moisture and intercepted rainfall for the consumptive pathway of evaporation (soil, surface, snow) [49,52].

#### 3.1.2. Blue Water Stakeholders

Water use, referring usually to blue water, is defined as its removal from its source and is distinguished into "withdrawal" where water returns to the water system by return flows or leakage, and as "consumption" [53], termed as "irretrievable or irrecoverable loss" [54] to signify that it may be transformed to a form not immediately returnable to its initial state, measuring the amount that is removed from rivers, lakes, or groundwater sources and evaporated to the atmosphere [55,56]. In [57], withdrawals and consumption are considered to be the maximum and minimum levels of scarcity. For the case of agriculture, water use may be defined to be the quantity of water which is in the process of evapotranspiration [58], two-thirds of which are supplied by plant transpiration [59,60] re-enforcing the role of green water. It should be noted that one of the end results of evapotranspiration, net precipitation on land (40,700 km$^3$/year), suffers a great deal of waste: 50% becomes floods and 20% is in areas that are too remote to be of immediate use [61].

According to the Organization for Economic Co-operation and Development (OECD) [62], in the case of water, "Stakeholders are herein defined as persons or groups who are directly or indirectly affected by water policy, as well as those who may have interests in it and/or the ability to influence its outcome, either positively or negatively" which includes "businesses depending on water for their process, those profiting from the water chain and those selling water dependent products" [62]. This is very close to Freeman's stakeholder definition of corporate stakeholder being "any group or individual who can affect, or is affected by, the achievement of a corporation's purpose" which includes employees, stockholders and customers [63] (p. vi, EXHIBIT 1.5 p. 25) and, most importantly, allows the OECD definition to be separated into immediate or primary and secondary stakeholders extending the secondary "potential users" by Newcombe [64] in terms of Freeman's definition to actual secondary stakeholders, seen also in [65] and in a similar way to a modernized version of [66] (p. 941, Table 1).

In Figure 5 the circular stakeholder diagram found in [54] is broken down into levels of impact, according to the nature of the stakeholders, from primary to tertiary, distributed according to normative decision. The immediate (primary) stakeholders are seen to be

agriculture, industry, and municipal water provision (which include household uses), while the whole population is a tertiary stakeholder.

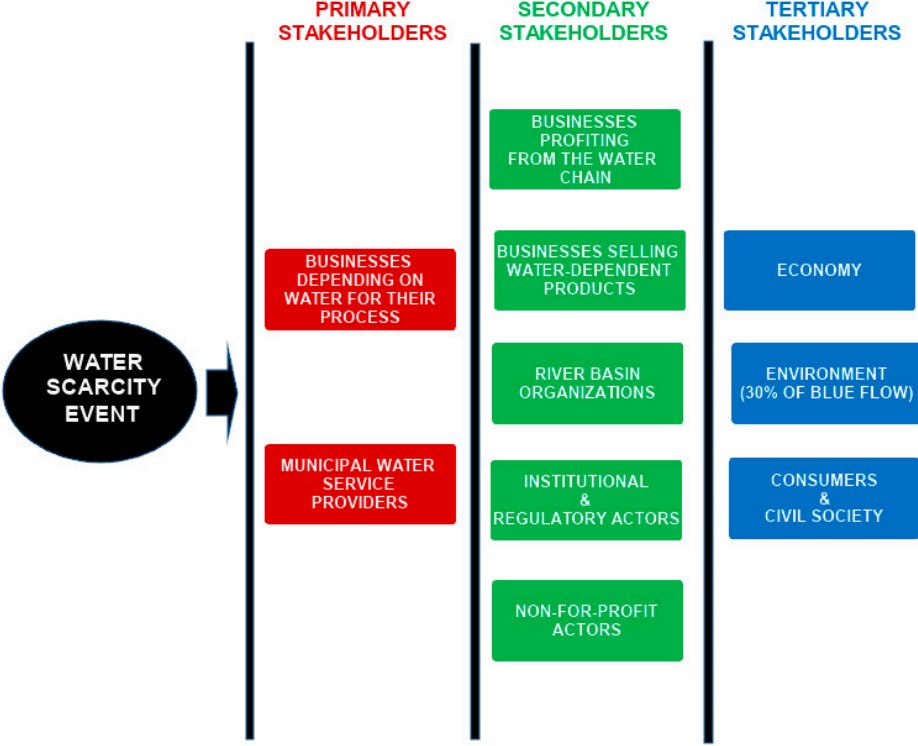

**Figure 5.** Major stakeholders (modified from [62]).

In Figure 6, agriculture is seen to be the dominant stakeholder due to the continuous pressure exerted by the increase in population and total consumption of food and the corresponding increase in land use land cover change.

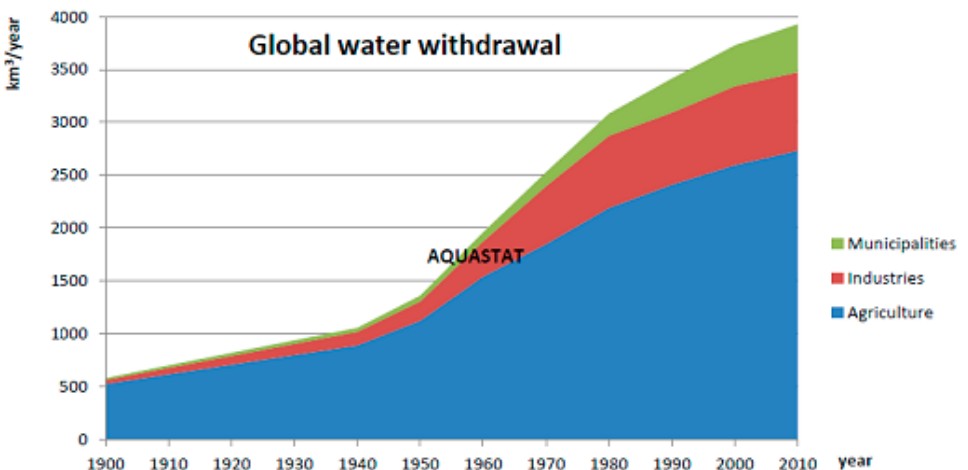

**Figure 6.** Global water withdrawal 1900–2010 (modified from [67]).

Withdrawal division by continent for primary stakeholders in Figure 7 shows disparate distribution for agriculture. This is due to population increases in Asia and Africa, which are not commensurate with those in Europe or the USA and Canada population component in the Americas. In addition, the fact that inferior technology is employed in Africa, leading to low yields per ha, should be taken into consideration.



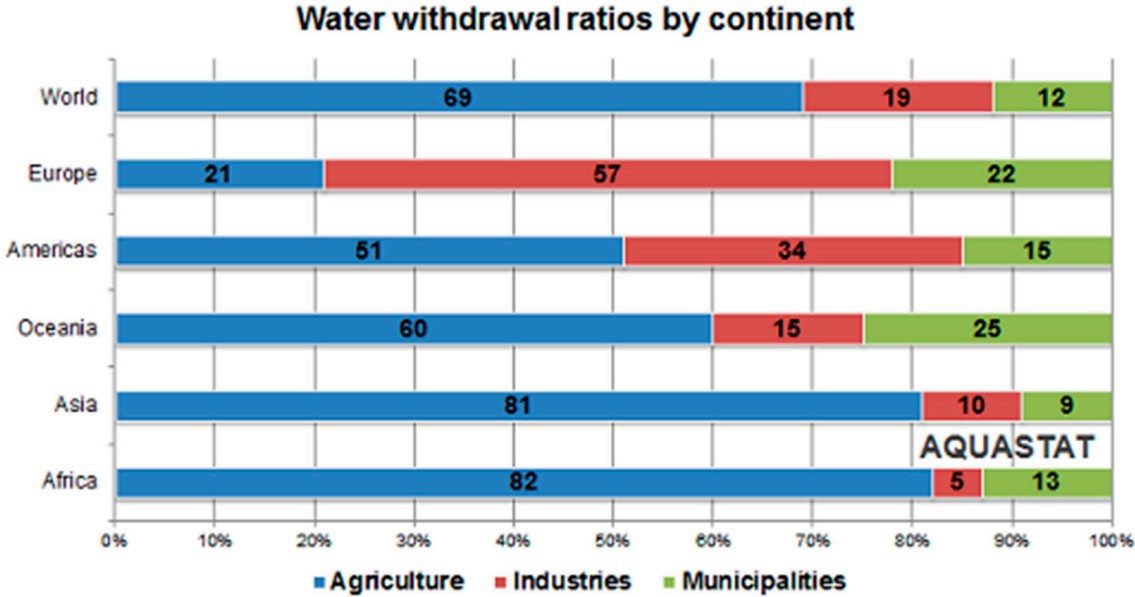

**Figure 7.** Water Withdrawal ratios by continent (modified from [67]).

Water withdrawal by country economic grouping (OECD, BRICS (Brazil, Russia, India, China and South Africa), ROW (Rest of the World)) are as below in Figure 8 and the differences between the OECD and the other two groups can be ascribed to the fact that BRICS' economic growth is sought in a more pressing and less rule-bound way than in OECD countries.

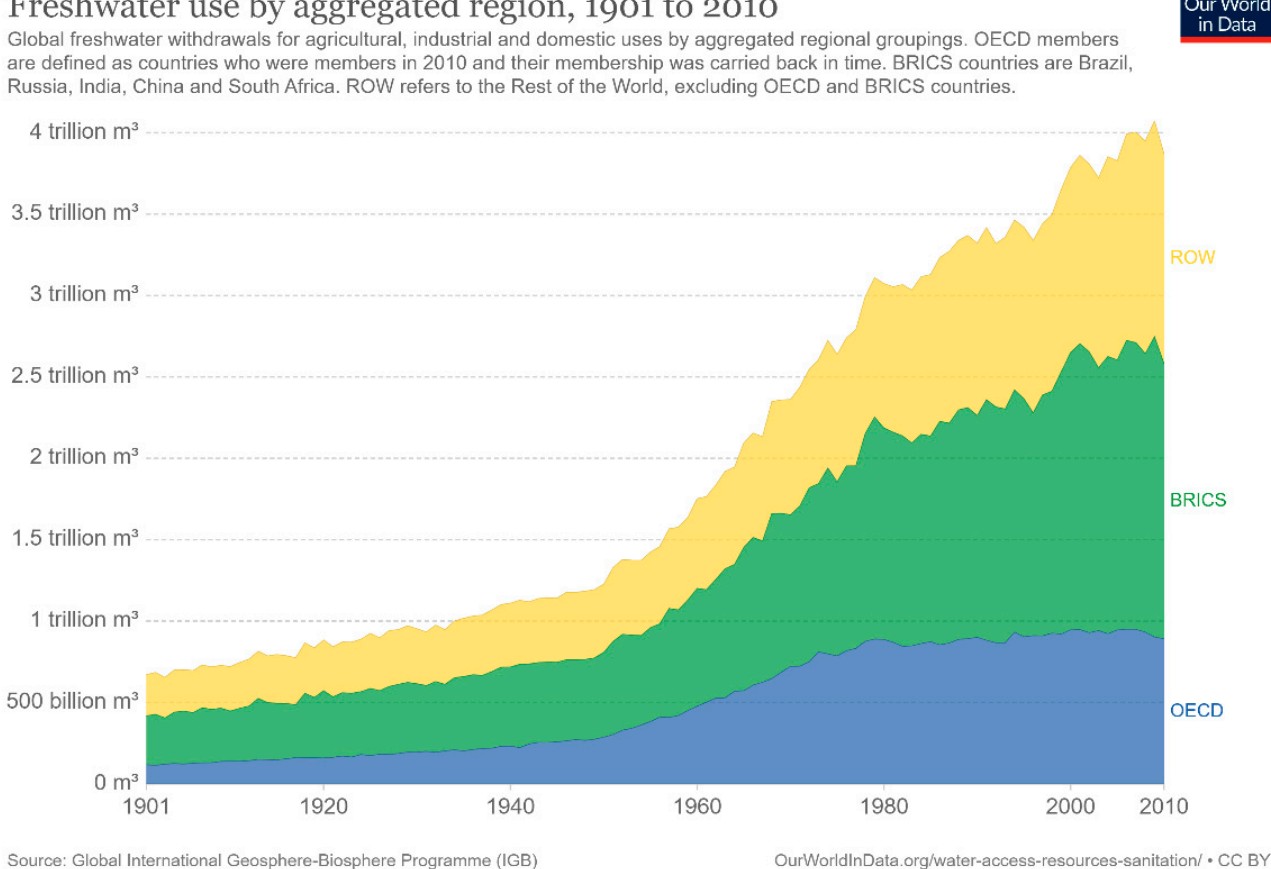

**Figure 8.** Water withdrawals by country economic grouping (modified from [68]).

### 3.1.3. Scarcity Definitions

Scarcity is an acceptable property of water as it is an irreplaceable input to goods that satisfy economic wants [69] and is, by its nature, a finite resource and essential good according to the Dublin Water First Principle "Water is a finite, vulnerable and essential resource which should be managed in an integrated manner" [1]. An initial institutional definition of water scarcity was given by Winpenny as "In popular usage, "scarcity" is a situation where there is insufficient water to satisfy normal requirements. However, this common-sense definition is of little use to policy-makers and planners. There are degrees of scarcity: absolute, life-threatening, seasonal, temporary, cyclical, etc. Populations with normally high levels of consumption may experience temporary scarcity more keenly than other societies accustomed to using much less water. Scarcity often arises because of socio-economic trends having little to do with basic needs. Defining scarcity for policy-making purposes is very difficult." [70] in FAO [71]. Using this as a basis, the 2007 U.N. definition, which stresses the relative nature of water scarcity, followed "The point at which the aggregate impact of all users impinges on the supply or quality of water under prevailing institutional arrangements to the extent that the demand by all sectors, including the environment, cannot be satisfied fully [...], a relative concept [that] can occur at any level of supply or demand. Scarcity may be a social construct (a product of affluence, expectations and customary behaviour) or the consequence of altered supply patterns stemming from climate change. Scarcity has various causes, most of which are capable of being remedied or alleviated." [72]. The exception to alleviation is defined by "absolute water scarcity" which is "wor to satisfy total demand after all feasible options to enhance supply and manage demand have been implemented" [73] usually set to less than <500 m$^3$ per capita of renewable water [74]. In addition, "chronic water shortage" exists when renewable water resources per capita are in the region 500–1000 m$^3$ and "regular water stress" in the region between 1000–1700 m$^3$ per capita [74]. It should be pointed out that the environmental flow requirement is predetermined as a mandatory 30% of the available sum of blue and green water [75] which reduces availability. However, water scarcity is claimed to be a "governance crisis, not a [water] resource crisis" [76].

Falkenmark distinguishes four modes of water scarcity [77]:

A.   Short-term growing season aridity
B.   Recurrent drought year intermittent droughts
C.   Soil degradation-induced landscape desiccation (man-made draught)
D.   Water stress induced by an exorbitant population number per unit of water cycle available.

According to the EU, water scarcity is interpreted as the case where "water demand exceeds the water resources exploitable under sustainable conditions" [78]

In modern settings, water scarcity may be defined in general as in [41]:

a.   an imbalance of supply and demand under prevailing institutional arrangements and/or prices [79,80]
b.   an excess of demand over available supply [81]
c.   a high rate of utilization compared to available supply, especially if the remaining supply potentials are difficult or costly to tap [82].

Simpler definitions are the case of where water resource availability is less hence the means of water availability and accessibility to the concerned population become restrained [83], when available resources are insufficient to meet the sum of total demand and minimum environmental flow [84] or the one where water is scarce for any reason [55].

Another definition of scarcity is based on the imbalance between supply and demand created by intra-annual and inter-annual fluxes which are manifested due to lack of freshwater natural or constructed storage, a fact that makes inadequate storage, i.e., storage scarcity, an equivalent definition to the classical definitions of water scarcity as it is operationally equal in terms of volume, location and timing to the supply-demand difference [85,86]. The theory stresses the point that all calculations should be based on physical parameter

quantification of supply and demand, and that, based on these results, decision making regarding additional storage should be a result of synergy between national and local authorities. Insufficient water storage exists in Canada (694 MAF), THE U.S.A. (1420 MAF), China (2280 MAF), India (245 MAF), and Pakistan (13.20 MAF) [87].

### 3.2. Blue and Green Water Scarcity

Water used in agriculture amounts to 70% of withdrawals [88] and 92 % of global fresh water consumption [89] which occupies 40 % of total available land divided into 1/3 for crop cultivation and 2/3 for livestock grazing [90]. The average annual blue water consumption is 1000–1700 km$^3$/year [91] while annual green water consumption on rainfed and irrigated cropland is in the order of 5000 km$^3$/year; to this should be added around 20,000 km$^3$/year for perceived to be managed grassland/grazing land and around 49,000–56,500 km$^3$/year for the support of non-agricultural ecosystems [52]. Irrigated agriculture (blue water) amounts to 20% of total cultivated land but yields 40% of total global produce [92], but 60% of total cultivated land is rainfed [55]. In year 2000 global cropland was $15 \times 10^3$ km$^2$ and pasture area was $34 \times 10^3$ km$^2$ [93]; while agricultural land was $46.8 \times 10^3$ km$^2$ in 1992, 36.763% of total land, and $48.6 \times 10^3$ km$^2$ in 2015, 38.177% of total land, an increase of less than 2% [94].

In general, virtual water, commodities, usually water moved in international trade in cereals [95,96], which by a wide interpretation of the Heckscher-Ohlin model are, in reality bundles of factors [97], including a sizeable amount of water, is exported by countries which do not suffer from either blue or green water scarcity, however, despite the fact that exports are mainly rainfed, there is a substantial increase in associated irrigation water expenditure, which beyond creating a depletion [98,99], is estimated to be a global increase of 17% in a 25-year period [100]. It is quantified as an "irretrievable or irrecoverable loss" in the sense of [59] for the exporting country and it should be taken into consideration that scarce water may be exported via virtual water trade, despite the fact that this increases scarcity at the exporting country [101].

### 3.2.1. Blue Water Scarcity

In the case of blue water, of which availability depends on climate (precipitation, temperature, radiation) and its variability, river flow directions, and the spatio-temporal distribution of lakes and reservoirs [102], upstream water consumption scarcity is attributed to shortage, the low availability of water per capita i.e., population driven scarcity. Stress is defined as high water use relative to water availability [103,104] as in the projected reduction of river inflows in the Tonle Sap Lake Basin, Cambodia [105], reductions in flow because of increasing temperatures in the upper Colorado River basin (UCRB) [106], multi-model global assessment [107], regional impact in Tangshan city, Hebei Province, China [108] and on the EU drinking water [109]. Blue water scarcity may be due to the competition over limited runoff and is usually measured as the ratio of blue water use to available blue water [43], a restrictive variant of the Water Resources Vulnerability Index [110], or the ratio of blue water use footprint to available blue water footprint [111]. According to Falkenmark, blue water shortage occurs on the basis of the number of people competing for a limited water resources, a Malthusian view based on a limited resource divided among an ever growing population, in which case "water crowding" is high when per capita water availability is less than the "principal water requirements" of 1700 m$^3$ cap$^{-1}$ yr$^{-1}$ [112] while below 1000 m$^3$ cap$^{-1}$ yr$^{-1}$ chronic water shortage could occur under certain conditions which may "be the single greatest and most urgent development constraint" [113] impacting 2.3 bn people [104]. In addition, South and Southeast Asia, under pressure of increasing agricultural production, have increased blue water withdrawals amounting to >60% of agricultural water use [114].

According to the planetary boundaries theory [115–120], where green water has not as of yet been incorporated [121], each main control variable is divided quantitatively at a global level into a safe operating space and a dangerously uncertain one

regarding withdrawals of any kind. While blue water available resources amount to 12,500–15,000 km$^3$ year$^{-1}$ [122,123], in total, the water withdrawal control variable has a safe operating space of 4000 km$^3$ year$^{-1}$, which, including a region of uncertainty, reaches to 6000 km$^3$ year$^{-1}$ [115]. In the case of blue water withdrawal as % of mean monthly river flow the total operating space, where the low bound is the safe part, is for low-flow months 25–55%, for intermediate flow months 30–60% and for high-flow months 55–85% [124]. It should be mentioned that water influences decisively all the basic control variables of this theory [125], hence global limitations on water may impose on the rest of the main control variables unnatural constraints and that in regional cases the limits imposed may be exceeded due to justified necessity [126].

Atmospheric evaporation recycling within drainage basins can reduce blue water consumption volumes by up to 32% [127]. A multi-dimensional diagnosis model (MDDM) assesses blue water at a regional level so that scarcity can be detected [128] and a review of the methodological challenges remaining for this assessment via footprint is in [129]. It may be that shifts toward highly resource-efficient cropping lead to increased demands of blue water if they are dependent on irrigation as is the case in China's Huang-Huai-Hai region [130], a manifestation of the Green Paradox [23].

In [131], the five elements of terrestrial water storage (TWS), among which are the constituents of blue water, groundwater and surface waters, are shown to be range bounded dynamic quantities trending below past ranges, in particular where groundwater is being withdrawn at an unsustainable rate while groundwater stress levels are defined and quantified using GRACE in [132].

### 3.2.2. Green Water Scarcity

Green water has three functions: regulatory (soil moisture, evaporation and transpiration flows which regulate via carbon sequestration and water as a component of greenhouse gas the planetary energy balance and climate system), productive (food, biomass and bioenergy production sustaining evapotranspiration) and overland water cycle regulation via moisture feedback evaporation [133]. It should be noted that green water flow is comprised of 59% transpiration, 21% plant interception, 10% floor interception, and 6% soil moisture evaporation [134], of whose partly contrasting roles are analysed in [135]. Green water is replenished by land precipitation or capillary rise from groundwater lying blow its zone [136]. Extreme precipitation redistributes its land impact by reducing the resultant green water and increasing soil erosion in comparison to regular precipitation [137]. Green water scarcity, measured by its fluxes as being approximated by its actual evapotranspiration [45], which is important in problematic regions e.g., >95% of sub-Saharan Africa is rainfed [114], where green water availability depends on agricultural area and its expansion [138], climate, in terms of rainwater partitioning, as in climate change effects on groundwater recharge [139], in recent advances in groundwater and climate change [140], in changes in mean and extreme precipitation in general [141] and over India [142] and Africa [143,144], in projected climate change for groundwater recharge in the Western United States [145], crop type as in the relationship of water scarcity with food production [102], is defined indirectly via the fraction of irrigation water required over crop water intake, a form of "green water deficit" [146], while alternative scarcity indicators are defined in [147], and the existence of scarcity is justified as there is a limit to green water resources in [148], where a supplement [149] is included, depicting numerical results for all countries. Assessment methods are e.g., in a case study in the Hai River Basin, China [150], by partitioning evapotranspiration into green and blue water sources [151], using GABI [152] and others. Green water scarcity, by calculating green water scarcity characterisation factors (CFs), occurs in Portugal and will get worse during the period 2046 to 2065 [153], the rapid development of urbanization may cause serious shortage of both blue and green water resources as is the case in rapidly-developing Xiangjiang River Basin in China [154]. Green water scarcity is induced in higher altimetry vegetation during a

temperature rise in lower altitudes as evapotranspiration in large areas over the Alps was above average despite low rainfall [155].

### 3.2.3. Blue and Green Water Interchange

Although blue and green water definitions might imply some form of definition-wise physical non-intersection between them i.e., the fact that that there is a flow path distinction and a *location* distinction [55], there exists physical interchange interaction between blue and green water in the ground where blue water moves downward into the ground according to the Dahl hierarchy (sediment scale(<1 m), reach scale (1–1000 m) and catchment scale (>1000 m)) [156], a form of percolation ending either in phreatic zone groundwater pockets, also seen in shallow unconfined aquifers [157], or flowing to create blue water flows [158], although, depending on the existence of semi-aridity conditions, there is transient resistivity to downward interaction as in the 200 km$^2$ Wailepalle watershed in the high Deccan plateau in India [159]. This interaction extends to the fact that terrestrial precipitation, which includes the blue water renewability package, has 65% of its source in terrestrial evaporation flows to the atmosphere [160], whose soil moisture component, like interception [161], is ephemeral (as seen in a simple model of the Budyko curve [162]). Additionally, there is interaction between blue and green water in terms of trade-offs, as seen in the Amazon [163] and via green-blue water accounting in soil water balance in [164]. Interlinkages also exist in the case of an arid endorheic river basin where hydrological cycling is heavily altered by human activities as is the case of an arid catchment in Northwest China [165]. Hence, they are not contiguously distinct either in situ or in composition, which poses a problem in measuring the two quantities unless the bilateral transference is explicitly assessed quantitatively.

### 3.3. Blue and Green Water Climate Interactions

### 3.3.1. General Climate Interaction

Climate, which is subject to global short term [166] and long term [167] changes [168] and variability (which can assessed on a millennial-scale, spanning 800,000 years and its influence on climate change [169] is due to the alteration of a large number of conditions) is depicted using the Köppen-Geiger classification [170]. The consequences are far ranging e.g., climate influences cleaner production [171].

Climate change is a statistically determinable change in the state of the climate which has a long-term duration due to natural (internal or natural variability)/anthropogenic causes [172]; climate variability, on the other hand, is the occurrence of annual distancing of main parameters from the long term mean as seen in Figure 3. At the global scale, in global climate models (GCMs), anthropogenically induced change overcomes internal variability at the decadal time scale [173] but, at regional and local scales they tend to be equally important, even during 50-year periods [174]. Natural variability, which at the regional level is distinguishable from anthropogenic forcing regarding precipitation [175], may be used in the detection of anthropogenic climate change as a cause or both regional and local precipitation [176] and in the case of Pakistan using the 1951–2015 APHRODITE dataset, a quantitative estimate of induced climate change variation in precipitation was deduced [177]. Climate models suffer from structural model uncertainty and parameter uncertainty [178]. As a general rule in ensemble forecasting (ensemble prediction systems (EPS)) [179], where ensemble spread is considered to be a proxy for prediction uncertainty, reliable predictions are described by the case where the time-mean ensemble spread about the ensemble–mean prediction equals the time-mean RMSE of the ensemble–mean forecast [180]. Hence, the more the forecasting systems used the less the resulting uncertainty, as in seasonal forecasting using the DEMETER data set [181], in flood forecasting using the TIGGE system [182] and the encouraging results shown in precipitation at sub-monthly time scales in [183]. Predictions at global and regional scale vary, as in [184], globally at 1.75 (1.65–1.85) °C increase over preindustrial temperature e.g., it is expected to have a global increase of heatwave frequency chance by 73 (67.2–77.9)% compared to 33% in the

1981–2010 period, global hydrological drought frequency chance will have an increase by 8.3 (6.1–9.9)% compared to 6.1% in the same period while global agricultural drought (SPEI) frequency chance will reach 23.6 (16.1–29.5)% compared to 9.4% in the same period. As global mean temperature increases to 4 °C, all positive or negative climate impacts become worse, while, at a continental level, these predictions vary. Regional climate models (RCMs) depend on global models by "regionalization" as in the review in [185], or by selection, as in [186], adding data to global models [187], perform better in EPS [188], and in weighted EPS [189] Regarding seasonal mean rainfall, coarse and high-resolution RCMs are in agreement if they are convection-permitting and the GCM is reliable [190].

Climate is a major water scarcity driver through its influence of global and regional interannual variabilities of precipitation as in the vulnerability of global water resources from climate change [191], climate change and water [192], climate change and changes in global precipitation patterns [193], regional and impact-related climate targets [194], the relationship between climate forcing and impact [195], and climate change at different levels of global temperature increase [184]. Short term variations are connected to long term ones [196] and in [197] regarding connection with seasonality. In terms of extreme rainfall, there is a link between these and temperature in tropical climates, where there is an increase in warm periods and a decrease in cold periods [198]. Additionally, climate-driven interannual variability of water scarcity impacts food production [199], in [107] regarding climate change on renewable water resources at the global scale in combination with population increase, in [200] regarding drought creating anthropogenic climate change, in [201] global assessment of the impact of climate change on water scarcity.

It should be pointed out that, in the case of precipitation models, including precipitation extremes, grid refinement plays a major role and general circulation models (GCMs) have been substituted by high resolution Regional Climate Models (RCMs) which labour under two problems, that of the requirement of specification of lateral boundary conditions as they impede self-consistent interactions between global and regional scales of motion [202] and that of the fact that simple resolution increases lead to very limited improvements in the long forecast range without model improvements, i.e., reconsideration of the physics picture [203]. Notably, high-resolution models perform better for East Asia than for India [204]. Blue water sources such as lakes influence climate through the carbon cycle [205,206] and modify the climates of their surrounding areas up to synoptic scale [207]. At the same time there is a definite trend of rapid and highly variable warming of lake surface waters [208], and a study of 20 Danish lakes showed for the period 1989–2006 where a surface water warming of ~2 °C and a cooling of deep water of ~1 °C were found [209]. The effect of the climate changes can be seen in [210] with respect to the Amazon and La Plata basins and in [211] regarding the Yarlung Tsangpo-Brahmaputra River (YBR) impacting assessed in China, India, Bhutan and Bangladesh, where in terms of RCP8.5 conditions, by 2035, the flow will increase by 12.9% (Bahadurabad, Bangladesh), 13.1% (upper Brahmaputra outlet) and 19.9% (Nuxia, China) relatively to the 1980–2001 period. In terms of precipitation trend due to climate change, the general guiding principle that, in the subtropics dry areas become drier and in the mid to high latitudes wet areas become wetter holds true [212], as seen in the transition to a more arid climate in Southwestern North America and in the trend of wet seasons becoming wetter and dry seasons becoming drier [213], and, in [16], climate caused changes in precipitation patterns have influenced total amounts of rainfall and extreme events (droughts and floods). In a warm climate, the anthropogenically caused increase in atmospheric water subject to precipitation leads to an enhancement of either moisture convergence or divergence increasing variability of precipitation water in a warmer climate enhances moisture convergence or divergence during wet or dry years, consequently increasing precipitation variability [214,215].

Additionally, in [216] adverse climate change will lead to reduced water availability in the countries that are already water scarce and to an increase in the variability with which the water is delivered. Climate variability impacts are found for both blue and green water and human water use affects regional climate [217], blue and green water

resources under CMIP3 and CMIP5 models [218], blue water in the Athabasca River Basin, Canada [219], groundwater [220], groundwater storage [221], river basins in the Western USA [222], impacts on hydrology and water resources in the Blue Mountains, Oregon, USA [223], in the making water resources in Phoenix, Arizona, vulnerable [224], and observed streamflow, evaporation, drought trends and water resources in the USA [225]. Impact of soil moisture-climate feedbacks on CMIP5 projections are shown in [226] and there is model agreement on forced response pattern of precipitation and temperature extremes [227]. In addition, 25% of exorheic river basins run dry without reaching the sea [228], and some are periodically dry, e.g., the Yellow river, the Colorado River, and the Ganges river [61], this being usually attributable to irrigation water withdrawals and associated evapotranspiration (ET) increases [229] due to aridity and semi-aridity [230].

As can be seen in [231], the most common climate type by land area is (14.2%, Hot desert) followed by (11.5%, Tropical savannah) and in the Köppen-Geiger Maps for the periods 1980–2016 and 2071–2100 there is a marked change in climate. At the same time some regions are chronically more sensitive to water withdrawals and availability than others [232], e.g., India, northern China, north and sub-Saharan Africa, the Middle East, and parts of Eastern Europe [233].

Besides flowing water, groundwater is climate influenced, as seen in [140] and in [234], in the central high plains aquifer, in strategic mid-latitude aquifers (such as in the Central Valley in California) and the aquifer beneath the upstream regions of the Indus River and Ganges River in Northwestern India, among others [235], as well as in climate-induced increase in pumping [236].

A closer look at some the world's more indicative regions follows.

### 3.3.2. Europe

In the EU (2012), with a total area of 4.3 M km$^2$, river basins make up an area of 987,914.5 km$^2$ (23% of total area) and suffer summer water stress. River basins with an area of 460,521.9 km$^2$ (10.7% of total area) suffer year-round water stress, while the corresponding projections for 2030 are 1,934,998 km$^2$ (45% of total area) and 1,288,885 km$^2$ (30% of total area), respectively [237].

While climate variability impacts blue and green water fluxes in Europe [238], in terms of RCM models, or otherwise, a rise in mean and extreme precipitation is projected for Northern Europe, as seen in a climate change simulation for Europe [239]; this attributing precipitation to changes in synoptic circulation [240], in an intercomparison of scenarios from regional climate models [241], in an ensemble of regional climate simulations [242], to an exploration of regional climate model projections [243], to human contribution and precipitation extremes [244], while the 20th century showed a precipitation increase in Northern Europe by 10–40% [245], and in an analysis of a high-resolution climate change scenario [246], as well as in a high atmospheric river (AR) contribution to precipitation [247]. In the Mediterranean region of Europe, the converse is projected, a reduction in mean precipitation with an increase in extreme values, as seen in an increase of extreme daily rainfall in the Mediterranean, but a decrease in total values [248], increase in precipitation in northern Europe and decrease in southern Europe is attributed to a poleward shift of the North Atlantic storm track [249], and there is low atmospheric river (AR) contribution of atmospheric rivers to precipitation in Southern Europe [247], which is confirmed by the fact that the first empirical orthogonal function (EOF) with S-PC 1, explaining 12.8% of total variance reflects the North-South contrast, which implies that oscillations are not "well-related" to the North Atlantic Oscillation (NAO) index [250]. In fact, that the contrast between the trends in Northern and Southern Europe may depend on the choice of index, e.g., it is more diffuse for S95pTOT than for R95pTOT [251], in simulation using a regional climate model [252], in regional climate simulations for Europe and the Alpine Region [253], while total precipitation will decrease over most of the considered domain from a high resolution double nested RCM simulation [254]. The effects of climate change can be seen in [255] regarding Portugal, while, in Northern Europe, the picture of changes in

extreme precipitation is approximately the same as that for the trend in total precipitation amount, and in Southern Europe, the same happens in winter, albeit slightly wetter in other seasons [256]. In [257], climate leads to a decrease in available water resources of >10 % in Central, Eastern and Southern Europe. Climate variability alters partitioning between the runoff sources and flow regimes in Swiss Rivers [258] and causes warmer summers to have more green and less blue water in the Alps [259]. For the period of 1979–2009, monthly precipitation trends of circulation changes seem to be of importance in Northern Europe in February and December, explaining wetting trends in Northwestern Europe in July, while, in the Mediterranean, February was dry relative to the rest of Europe [260].

### 3.3.3. Africa

Of particular interest is Africa in general, despite the fact that in Figure 6, water scarcity is mainly attributed to economic scarcity, as precipitation is due to deep convection, which is the result of a hydrodynamic instability in the troposphere [261], climate change will amplify existing stress on water availability [262]. Its 2015 population of 1.19 billion has a median projection of increasing to 1.68 billion people in 2030 (42% increase) [263], and the problematic sub-Saharan region, in particular, with a 2015 population of 995.5 million [264] and 2030 projection of 1.4 billion [265] (40% increase). In addition, climatic changes impact river runoff, increase blue water demand. and increase the risk of shallow groundwater contamination via intense rainfall [225]. Evaporative losses in the Zambezi are increasing [266] and, in [257], climate leads to the decrease of available water resources by >10% in Okavango and Limpopo in Southern Africa. In addition, the tropical mechanism is such that if the air above the Indian Ocean boundary layer warms up it will impact Africa by reducing the local precipitation, leading to hydrodynamic stability, which results in a reduction in precipitation [267].

In general, the variability of interannual rainfall is high, as shown in [268], where the Africa Rainfall and Temperature Evaluation System (ARTES) was employed, especially in the Sahel region [269]; and a new concept had to be introduced, that of near-surface storage, to address evaporation losses from rain falling on dry soil [270]. This was shown in a study in Nigeria where, out of rainfall, only 12% becomes green water while 70% evaporates without penetrating the soil to such a depth where it could become green water [271]. Decreasing stream flows for rivers in Sudan and increasing discharge for those in the Sahel were found in [272,273]. In terms of groundwater, the predictions are that recharge will increase in Sahel and decrease in South-West Africa [274] and a review of these estimations for the entire continent is presented in [275]. In [276], climatology, annual cycle, and interannual variability of precipitation and temperature are simulated over West Africa, and climate variability impacts are shown for west African rivers [277]. In [278], Sahel, West Africa (WA) and Southern Africa (SA) are identified as CMIP5 type climate change hotspots, and there is oceanic forcing of precipitation in the Sahel [279], as well as that historical analysis predicts substantial drying over much of the Sahel and East Africa during the primary growing season by the end of the century [264]. It should be noted that there are about 80 river and lake basins in Africa, 21 of which are used by over 10 sovereign states [280] and misuse of upstream water privileges could lead to an "water war", a conflict centered on water scarcity and trans-boundary water sources [280,281]. In general, future trends lead to that Mediterranean Africa and the Northern Sahara will suffer a decrease in annual rainfall, which will intensify at the Mediterranean coast and in southern Africa's winter rainfall region. It will increase in East Africa while the Sahel rainfall increase will be balanced out by evaporation [282]. Regarding precipitation projections over the Democratic Republic of Congo, changes in both frequency (RR1) and daily mean intensity (SDII) lead to a tendency towards less frequent but more intense precipitation, while in Botswana, Zimbabwe and Mozambique, RCMs project a robust decrease in both mean precipitation and frequency (RR1), with a consequent increase in the number of consecutive dry days (CDD), up to more than 12 days/season. In Somalia there will be an increase in

annual and SON mean precipitation, together with an increase in both maximum daily intensity (RX1 day) and frequency of extreme events (R10 mm) [283].

### 3.3.4. Asia

In [222], projected precipitation changes over the south Asian region for every 0.5 °C increase in global warming are shown. In [257], it is shown that climate leads to a decrease in available water resources by >10 % in the Zhu Jiang catchment in southern China, in [284], future rainfall events are likely to be more intense, leading to run-off water losses and rivers in South Asia are likely to exhibit decreased summer flows (after an initial increase) and increased winter flows, which leads to the necessity of increased storage facilities. In [285], climate induced decrease in the summer monsoon rainfall in 2009 caused the most severe drought experienced in Southeast Asia since 1875, and exceptions from normal years in terms of drier and wetter years, and records for the highest and lowest temperature were observed around the globe during the 2000s. Climate variability impacts on both blue and green water in the Upper Ganjiang river basin in China [286], the Erhai Lake Basin of Southwest China [287], the Taihang Mountain Region, China, over the past 60 years [288]. Flows under natural conditions in inland river basins in Northwest China are covered in [289], blue water in India in [290], green and blue water over Asian Monsoon Region in [291], and blue water from snow and glacial melt for Asian river basin hydrology in [292]. In South Asia rainfall intensity has increased but the number of wet days has been reduced [293]; thus, increasing blue water consumption. Under RCP4.5 and RCP8.5 scenarios, using an increase of 1.5–2.5° C, daily precipitation extremes could increase by 4 to 6 times over India, while annual mean precipitation would be insignificant at the 1.5 °C level. Regarding Bangladesh, Bhutan, India, Nepal, Pakistan and Sri Lanka, by the end of the twenty-first century the country-averaged annual mean precipitation is projected to increase by 17.1% (2.2–49.1%), 18.9% (−4.9 to 72%), 27.3% (5.3–160.5%), 19.5% (−5.9 to 95.6%), 26.4% (6.4–159.7%), and 25.1% (−8.5 to 61.0%), correspondingly by the end of the twenty-first century under the SSP5-8.5 scenario (uncertainties in parentheses whose size speaks volumes) [294]. From the 50s to the early 2000s, Southern Vietnam, the northern part of Myanmar and the Visayas and Luzon Islands in the Philippines see heavy precipitation increases and northern Vietnam sees decreases [295]. Over South Asia, extreme precipitation occurs mainly during the summer and autumn, accounting for more than 40% of the total precipitation in winter over India and it occurs during all seasons over Southeast Asia, exhibiting a decline in the autumn and a maximum in winter [296]. Aridity is expected to increase in Central Asia, along with high temperatures in summer and fall, and decreased precipitation, particularly in the western regions of Turkmenistan, Uzbekistan, and Kazakhstan [297]. A comparison of Asian precipitation between 1920's and 1990's is shown bellow.

### 3.3.5. WANA Region and the "Arc of Crisis"

The Western Asia and North Africa (WANA) region (Algeria, Morocco, Tunisia, Libya, Egypt, Eritrea, Ethiopia, Sudan, Turkey, Cyprus, Iraq, Israel, Jordan, Lebanon, Syria, Iran and all countries in the Arabian Peninsula, Pakistan and Afghanistan), which is, in essence, the IMF MENAP region [298], has the lowest per capita renewable water resources [299] and includes, in part the "Arc of Crisis" (Somalia, Sudan and Egypt, Yemen, Iraq, Pakistan and Afghanistan) [300]. Leading to 2025 (base year 1995), the WANA region is expected to continue over-pumping, with SMAWW (surface maximum allowed maximum water withdrawal) and GMAWW (groundwater maximum allowed water withdrawal) increasing at annual rates of 0.66% and 0.12% respectively, while population will increase by 66.1%, GDP/capita by 88%, irrigated area by 18.4% and livestock production by 87%, all of which influence water supply and demand [301]. Algeria has an estimated 30% rain deficit, impacting watercourse flow regimes, siltation reduced dam storage capacity by 2 to 3% per year and groundwater level dropped below 20 m [302]. Due to climate variability the country's center, west and some province of the east are the most vulnerable to the

upcoming (2027) water resources deficit [303] while acute climate sensitivity is expressed with a hydrological water stress index (HWSI) of 33.5 and a reversed water poverty index (RWPI) of 47.9. The semiquantitative evaluation of temperature changes in absolute terms ($\Delta T$) are in the bracket 0.5 °C < $\Delta T$ < 1 °C [304] and all model simulations predict expansion of the desert climate zone at the expense of both temperate and steppe climate zones [305]. Somalia, taking into account annual average rainfall pattern, has a sub-humid to desert climate, two rainy seasons and two dry seasons, and the climate is influenced by an inter-tropical convergence zone (ITCZ) and the Somali jet [306]. It has an estimated available water of 14.7 km$^3$ with an annual withdrawal rate of 3.3 km$^3$ and water ownership belongs to the private sector, where high prices are imposed [307]; there is a state of perpetual armed conflict, and half the primary water sources are serviceable and 2.7 million people are in need of humanitarian aid, which includes the need for water [280]. Yemen's water supply was about 1100 m$^3$/capita/year in the 1960's, near the water poverty line; by 1990 it dropped to 460 m$^3$/capita/year. Water gathering using standard fog collectors (SFC) was tested in Hajja in 1989 and found to be promising [308]. By 2012, water supply dropped further to 120 m$^3$/capita/year while the current national population growth rate is about 3.5% annually and over two million Somali immigrants have been accepted as refugees [309]. In Sana'a, Yemeni farmers increased water well depth by 50 m over a 12-year period, but despite this, the extracted water was diminished by 66.6% [300]; the water table is declining in average by about 6–7 m annually due to groundwater over-abstraction [310] and fresh water withdrawal/available freshwater resources was 170% in 2014. Egypt suffers from explosive population increase, from 90 million in 2015, going on to 140 million in 2037 and 170 million in 2050 which correspond in Nile water/capita/year, to 611 m$^3$/capita/year, 392 m$^3$/capita/year and 324 m$^3$/capita/year respectively [311]. The incoming volume of Nile water is constrained by treaties with Soudan [312] and by the Grand Ethiopian Renaissance Dam (GERD), where filling the dam will disrupt the flow into the Egyptian part of the Nile and the post-filling period might include a severe multi-year drought [313]. Pollution is highly problematic [314] but the main long-term dangers arise from climate interaction, seawater rise threatening the Nile Delta [315], climate variations and change to the Nile in terms of climate and hydrology of the Upper Blue Nile River [316], as well as impacts from climate change on Blue Nile flows [317], sea-level rise and climate change impacts on the lower Nile delta [318], and, in the future, hot and dry years will worsen Nile Basin water scarcity [319]. Increased water requirement due to higher temperatures will occur as evaporation from the High Aswan Dam is over 10% of the Nile flow [320]; an increase by 3–3.5 °C may be manifested by 2060 [321,322]. Climate change affects Arab countries [323] and the work in [257] shows that climate leads to the decrease of available water resources by >10% in the catchments of the Euphrates/Tigris in the Middle East, a decrease in Syrian precipitation from a combination of natural variability and a long-term drying trend [324], while climate variability will impacts the water resources of the Greater Zab River, Iraq [325]. In the cities of Riyadh, Jeddah, Mecca, Medina, Al-Ahsa, Ta'if, Tehran, Mashhad, Isfahan, Karaj, Tabriz, Shiraz, Qom, Ahvaz and Baghdad a baseline water stress >80% is expected [326].

*3.4. Action at a Distance*

3.4.1. Climate Teleconnections

However, teleconnections, "statistically correlated climate-related patterns between remote geographical regions of the globe" [327] or "a cause-and-effect chain that operates through several intermediate steps and leads to a linkage between two parts of a system" [29], influence precipitation, as in the case of the upper Medjerda Basin [328], the modulation of ENSO-precipitation teleconnection by the interdecadal pacific oscillation [329], across the combined North American, monsoon Asia, and Old World drought atlases [330], predictability of winter precipitation in Southwestern US via interhemispheric teleconnection [331], the case of water level regime of selected Polish lakes [332], the global influence of the ENSO-Indian monsoon [333], and, therefore, it may be considered that

part of local precipitation is indirectly "imported" or "curtailed" as a result of the imported change of the conditions influencing its quantity. Precipitation-shed algorithms [334] lead to the computation of teleconnectional spatial dependence for any specified rainfall portion over any specified area. The most well-known case of influence via teleconnection across the globe impacting at a regional scale is the El Niño Southern Oscillation (ENSO) via temperature on precipitation [335] and a quantification of the probability of the occurrence below normal, near normal, and above normal values of precipitation and near-surface temperature in relation to ENSO is presented in [336].

The distance of teleconnection influence may rise to thousands of kilometres e.g., Upper Blue Nile Basin rainfall and flows are influenced by the El Niño Southern Oscillation (ENSO) as in link to the Blue Nile River Basin hydrology [337], climate teleconnections and water management [338], summer rainfall over the source region of the Blue Nile [339], influence on the natural variability of the Nile River [340], and on precipitation and surface temperature over the Upper Blue Nile Region [341]. Influences extend to Caspian Sea level variability [342], as well as the west and northwest of Iran, west coasts of the Caspian Sea and the Southern Alborz Mountains [343]. They also extend to seasonal precipitation [344] and summer climate in China [345], to Mediterranean precipitation variability [346], European winter precipitation anomalies [347]. The North Atlantic Oscillation Index (NAOI) extends to Lake Windermere [348]. Additionally, the warming of the Indian Ocean is expected to disrupt rainfall in Eastern and Southern Africa, increasing undernourishment by 50% by 2030 [349]. Groundwater is subject to teleconnection influence such as in non-stationary groundwater level response [350], groundwater level response in U.S. principal aquifers [351], as well as climate forcings on groundwater resources of the USA's West Coast [352]. ENSO is also influenced by drought as in projections of future groundwater drought [353], drought and climate teleconnection [354], in Iran [355], China [345], as well as a series of droughts in the USA in 1988 [356], Further works address the turn of the century [357], droughts in general [358], in mainland Southeast Asia [359], in Northern Chile [360] and East African drought during rainy seasons [361]. Flood occurrence is influenced as well shown by the 2010 Pakistan flood and the Russian heat wave [362], as well as in Iran's Kan River basin [363], in the Southern Great Plains [364], the Missouri River Basin [365], the Yangtze River [366] and on a global level [367]. There is also extreme precipitation in North America [368], Central-Eastern China [369], over China in general [370], as part of the ENSO asymmetric effect [371], and in Northern South America [372]. Extreme precipitation relationships in the Mediterranean region exist [373], as well as variability of extreme precipitation over Europe [374] and on variability analysis of extreme precipitation in Turkey [375].

### 3.4.2. Evaporation and Moisture Import/Export

Based on work in [376,377] on the recycling ratio (RR) of the two-dimensional extension of the Budyko model [378], and its variants/extensions (also in [379] and a review of models in [380]), the fraction of precipitation over a defined region that originated as evaporation from that same region led to an assay at the national level of evaporative sources and sinks [381,382]. This was an estimation of how much precipitation was due to local and non-local evaporation and the determination of the nationality of non-local evaporation based on the classical equation, $P = P_L + P_A$, where the left side ($P$) is total precipitation and the right side is the sum of precipitation originating from local evaporation ($P_L$) and on advected evaporation ($P_A$). The method used was a quasi-isentropic back-trajectory (QIBT) scheme [383,384] which was performed on global gridded precipitation data found in [385] employing multiple tracers of atmospheric moisture [386], of which the paths were calculated using reanalysed fields of winds and temperature from [387]. A list of countries and their import-export properties are presented in [382] and a connection is shown by the result that, under certain conditions, water vapor is exported from less humid countries to more humid ones. Similar work was compiled for Central Europe, the Balkans and Spain [388], Colombia [389], atmospheric rivers over the West Coast of the United



States [390], for summer rainfall in the Southwest United States [391], in China [392], in the connection of water sources and precipitation recycling in the MacKenzie, Mississippi, and Amazon River basins [393], and in the Orinoco basin in Equatorial South America [394].

Terrestrial evaporation import/export, where in the case of import, advective precipitation is produced [395], is based on the notion of 'precipitationshed', "the upwind atmosphere and upwind terrestrial land surface that contributes evaporation to a specific location's precipitation (e.g., rainfall)" [396]. This is a form of teleconnection which may go down to smaller distances under particular conditions, which in essence is a land–sea area, enclosing the source of terrestrial evaporation surrounding the sink which receives the precipitation. Physically it is moisture recycling which returns via the atmosphere as downwind precipitation [397]. Western Sahel, Northern China, and La Plata are considered to be sink regions dependent on land moisture imports [334]. Forty percent of precipitation in certain Eastern Africa arid regions are connected to irrigation agriculture in Asia [398] and influences precipitation in the Amazon Xingu Basin [399]. In addition, 19 out of a total of 29 megacities were found to be dependent, in excess of 33%, on the precipitation shed mechanism for water, four megacities were already suffering from a series of problems [400]. Karachi (also suffering from supply problem, contamination, revenue recovery, industrial pollution and climate change), Shanghai (also suffering from pollution, salt water intrusion, the influence of major hydraulic projects and flooding), Wuhan and Chongqing (also suffering from pollution and wastewater treatment), were found to be highly vulnerable to the land change of the source component [401].

### 3.5. The Direct Human Intervention Scarcity Driver

Another major water scarcity driver is direct human intervention (HI) (land use and land cover change (LULCC) [402,403], man-made reservoirs for electricity generation, increase of wealth and human water use due to population increase [404]), which increases water scarcity for 8.8% (7.4–16.5%) of the global population, which is downstream but decreases it for 8.3% (6.4–15.8%) of the global population which is upstream [405]. Human intervention in terms of dams, due in part to increased surface evaporation [404], and water withdrawals, is seen to be have a more powerful impact than the climate [406]. LULCC also is directly related to population growth [407]. The anthropogenic greenhouse era may have begun thousands of years in the past, and not 150 to 200 years ago during the Industrial Revolution, as is the going hypothesis [408], and, in terms of net photosynthetic accumulation of carbon by plants, which is defined as net primary production (NPP) consumption, this has doubled in the 20th century [409]. This has led to a 31–32% consumption of the total amount of NPP generated on land [410,411]. Also, in [412] for total 2009 NPP prediction of 50.05 Pg C, an increase of +0.14 Pg C during the period 2000–2009. In [413], human activity induced climate caused temperature increase, leading to water resource reduction, which was the same in [414] and consequences are analysed in [415]. In 2016, human water use via the consuming (producing) of primary and manufactured goods and services from "primary crops and livestock", "primary energy and minerals", "processed food and beverages", "non-food manufactured products", "electricity", "commercial and public services", and "households" sectors accounted for 33% (91%), ∼0% (1%), 37% (<1%), 13% (1%), 1% (2%), 15% (3%), and 2% (2%) respectively, of the world's total blue water consumption [416]. In addition, according to the planet boundary theory, human impact is amplified by its interaction with the Earth system [417].

### 3.5.1. Land Use and Land Cover Change

During the period 1700–1990, cropland increased from $2.6 \times 10^6$ km$^2$ to $14.71 \times 10^6$ km$^2$ (565% ↑) and pasture land increased from $5.24 \times 10^6$ km$^2$ to $34.51 \times 10^6$ km$^2$ (658% ↑); the world population went from 605.4 million to 5301.8 million (875% ↑) [418]. During the period of 1700–1989, irrigated cropland went up from $0.08 \times 10^6$ km$^2$ to $2 \times 10^6$ km$^2$ (2500% ↑), grassland/pasture remained approximately the same, and during 1700–1983 forest/woodland dropped from $61.51 \times 10^6$ km$^2$ to $52.37 \times 10^6$ km$^2$ (14.86% ↓) [407].

It should be pointed out that the availability of land resources and local natural conditions [419] impose a nominal limit of a total cropland availability, in the range of $2.7 \times 10^9$–$3 \times 10^9$ ha [420,421], half of which is already cultivated [422]. Cropland expansion is virtually impossible without deforestation [423], a climate driven calculation which places it in the region of $300 \times 10^6$ ha to 2050 [424].

Land use land cover change (LULCC) reshapes water provision [425], the resulting deforestation raises the water table and, if the underlying layer is problematic, e.g., salty as in the case of Australia, it results in soil destruction and, in general, it mediates the trade-off of ecosystem services depictable in the three dimensional framework of (space, time, reversibility) [426], in particular of freshwater ecosystem services (FES) as in the case of Bangladesh, where long term reduction is traced to LULLC [427]. Moreover, deforestation reduces the flow of green water to the atmosphere, which increases the time needed for moisture recycling. However this is balanced out by evaporation from compensating increased irrigation demand [428]. LULCC is caused by biophysical constraints and potentials, economic factors, social factors, irreversibility and uncertainty, spatial interaction and neighbourhood characteristics and spatial policies at national and subnational level [429]. At a global scale, proximate LULCC drivers are agricultural expansion, urban growth, infrastructure development and mining, which are responsible for 80% of deforestation [430], while, in another study for the period 1990–2008, 46% went for livestock pasture, 11% for crops for animal feed, and the remaining 43% for agriculture [431]; at the global-continental scale climate, freshwater availability and soil are the drivers behind land use patterns [432].

In addition, "land-grabbing" type investments are a cause of LULLC e.g., in Africa there were 84 deals of 100,000 ha or more, out of 190 at a global scale, of which 2 deals in Sudan and the Congo Republic exceeded 1,000,000 ha [433]. These diminish the availability of blue water to others, since 18% of these (91,000 ha) require more than 50% of water from blue water sources [434]. In the tropics, agriculture is the main driver [435], while in developing countries commercial agriculture accounts for 40%, subsidence farming for 33%, mining for 7%, urban expansion for 10%, and infrastructure for 10% [436]. Additionally, the main drivers for deforestation in Argentina, Bolivia, Brazil, Paraguay, the Democratic Republic of the Congo, Indonesia, Malaysia and Papua New Guinea, in the period of 2000–2009, were beef, soy, palm oil, and wood products [431]. LULCC affects the climate via regional energy fluxes, impacting precipitation trends [437] and temperature [438] and, hence, intervenes actively in the hydrological cycle [439]. Its impact on climate is on local [440,441], regional [442] and global scale [443,444], may dampen or enhance the impacts of increasing $CO_2$ [445] with temperature consequences, is a main cause of soil degradation [446], which may lead to increased LULLC, alters the planetary boundary layer (PBL) structure by enhancing the vertical movement of air [447] and, if the area is large enough, it affects remote areas' rainfall via teleconnection. This is shown in an assay of the influence of land-use change and landscape dynamics on the climate system in [448], in hydro-climatological teleconnections resulting from tropical deforestation [449] and the impacts river flows as well [450], reductions in run-off as in the Guishui River Basin, China (where a 5% reduction was found) [451]. Land–atmosphere coupling induces climate change in Europe [452]. In [453] during the 1953–2001 period, expansion exceeding 5% significance in the areas of arid ($4.2 \times 10^5$ km$^2$ decade$^{-1}$) and continental climate ($2.3 \times 10^5$ km$^2$ decade$^{-1}$) north of 55° N and shrinkage of polar climate ($-2.9 \times 10^5$ km$^2$ decade$^{-1}$) and continental climate ($-3.2 \times 10^5$ km$^2$ decade$^{-1}$) south of 55° N was found. In non-Amazonian South America, where the environment is semi-arid and the population is around 200 million, LULCC is already causing water stress and reduced agricultural productivity as more than 3.6 million km$^2$ (58% of their potential natural vegetation) has been lost and may impact the Central Andes and Chilean Matorral where a weaker hydrological cycle is projected along with increased risk of lower water availability [454]. Similarly, in the Brahmaputra Basin, using a SWAT model with calibration parameters, such as surface runoff, groundwater, snow, ET, and the routing process for the basin's hydrology, it was found that a LULCC to agriculture scenario of 72% by 2070

would shift precipitation from the monsoon months towards the winter, thus increasing drought risk during early monsoon months [455]. In the case of the Jedeb mesoscale catchment, Abay/Upper Blue Nile basin, Ethiopia LULCC during 1973–2010 caused the soil moisture condition parameters to follow a gradual decreasing trend, increasing surface runoff in terms of high flow by 45% in the 1990–2000 decade while low flows decreased by 15% in 1970–1980, 39% in 1980–1990, and up to 71% in 1990–2000 [456]. A study on annual surface runoff and evapotranspiration was done, in the same general region, for the drought prone watersheds of Kasiry (highland), Kecha (midland), and Sahi (lowland) for 1982–2016/17, which showed runoff increases from 4% in Kecha to 28.7% in Kasiry and evapotranspiration ranged from 15.8% in Kasiry to 32.8% in Kecha despite climate variability induced evapotranspiration increase, ranging from 33.6% in Kecha to 42.1% in Kasiry [457]. Global land cover annual maps for the period 1992–2015 are in [458].

Sub-Saharan Africa, during the period 1975–2000, showed (using a stratified sampling strategy) a 57% increase in agriculture area, from 200 Mha to 340 Mha, at the cost of a 21% loss of forest and non-forest natural vegetation per year; in total, 131 Mha and an increase in barren land by 15% (6.5 Mha) [459].

However, climate (temperature) and precipitation sensitivity are an issue in LULLC e.g., the southern provinces of Canada, northwestern and northcentral states of the the United States, Northern Europe, the Southern Former Soviet Union and the Manchurian plains of China, are temperature-sensitive while the Great Plains region of the United States and Northeastern China are precipitation sensitive [460]. A strong relationship is observed between temperature and LULLC in general and between rainfall and LULLC in Southeast Asia in particular [461], where under RCP4.5 LULCC accounts for <10% of the projected temperature rise, averaged over sub-regions, but at the local scale may account for up to about 30% [462]. In South America, local impact of LULLC changes mesoscale circulation patterns, increasing natural vegetation productivity by 10% in the northwest and decreasing it in the southeast [463]. Sensitivities of surface temperature to LULLC, which induce biophysical changes, are scale-dependent due to atmospheric feedbacks [464].

### 3.5.2. Population Growth

Water scarcity is also caused by high-population pressure under the conditions of limited water availability [465], as according to the interpretation of the Ehrlich-Holdren equation, $I = PAT$, where $I$ is the negative impact on our life-support systems caused by our species, $P$ is the population size, $A$ is consumption per capita and $T$ is the measure of the technology servicing and driving consumption [466] (p. 58) and holds true only for a homogeneous country-wide study [467]. An increase in population is bound to increase water consumption, and increasing demand for water in this way outbalances the effects of global warming and is directly related to population growth [191]. In [6] it is pointed out that population growth is responsible for the increase of both water and agricultural production. Further, population is directly connected to climate, such as under the condition of constant productivity and technology. In the long run, global temperature is logarithmically related to population, if population growth is constant then temperature may be linearly related to it [468], as well as to land human carrying capacity, defined as "the maximum population that can be supported at a given living standard by the interaction of any given human-ecological system" [469] (p. 121). A modern theory of population dynamics takes into consideration agricultural production, the crop prod. Index, and access to water, as seen in [470]. Notably, renewable internal freshwater resources per capita (cubic meters) went from 13,403 in 1962 to 5933 in 2014 [471], a 55.73% drop, while world population went from 3.125 billion in 1962 to 7.254 billion in 2014 [472], a 56.92% increase and water consumption went up from 1840 billion cubic meters in 1962 to 3990 billion cubic meters in 2014, a 53.7 % increase. This demonstrates a more or less linear type relationship between renewable internal freshwater resources, which are shown to be relatively constant, population and water consumption, which are demonstrably increasing. MENA population went up from 138.47 million in 1962 to 417.9 million in 2014 [473], an

increase of 201.7%, which is expected to reach 692 million by 2050 [474]. Water demand will go up by 500% from 2010 levels [475], and this country set has the highest regional deviation from average annual surface water availability (70%) and water withdrawals as share of surface water availability (>80%), with >80% of total water withdrawals going to agriculture [476]. Future projections are 9.2 billion by 2050 [477], 9.7 billion by 2050 and 10.9 billion by 2100 [478]. It should be pointed out that climate-induced migration with major triggers drought and desertification [479], e.g., as in the cases of Bangladesh, Ghana, Ethiopia and Sudan [480], and also conflict, e.g., migrations to and from Somalia [481,482] plays a role in population distribution outside the birth-death actuarial projections.

Indicatively, from 1950 to 2010 the relationship between global population and water withdrawal is manifest, the difference in water for agriculture growth being ascribed to intertemporally increased productivity, as seen in Figure 9.

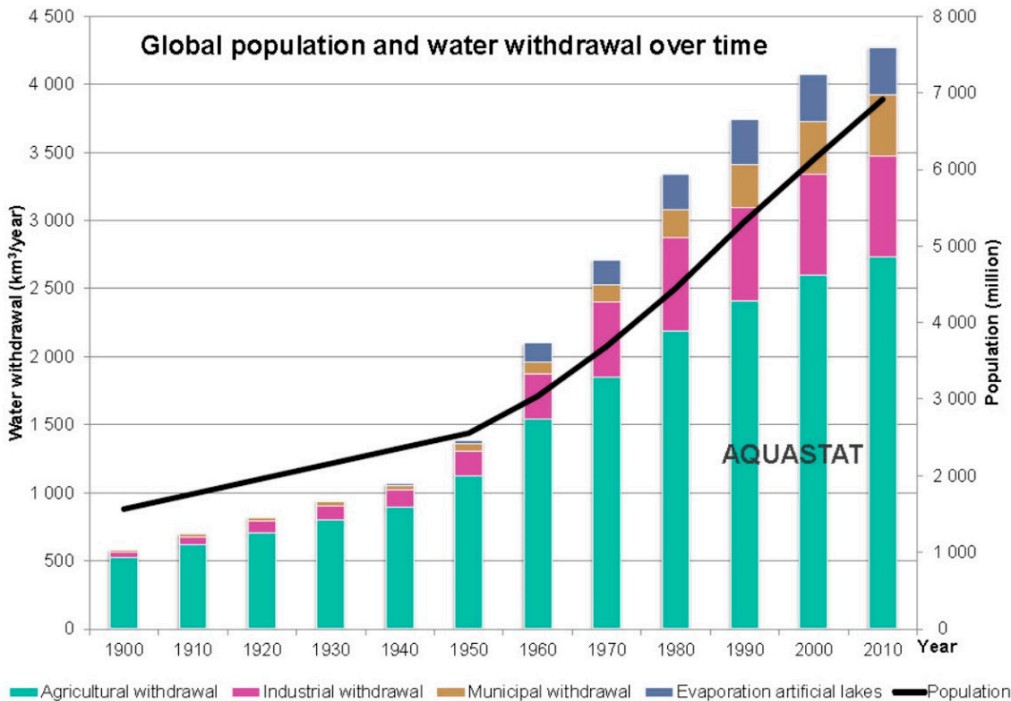

**Figure 9.** Global Population and Water Withdrawal 1900–2010 (modified from [483]).

Regarding population statistics to date, Africa seems to be moving towards some hybrid form of Shared Socio-Economic Pathways (SSPs) [3] (p. 19), scenario SSP3 in terms of population (comparatively high population growth), to the extent that, by 2100, 40% of the world's population will be in Africa [478] while the other continents are in a slow variability mode or declining, e.g., Europe and Asia.

The relationship between water and population has been discussed and analyzed extensively, e.g., by Falkenmark et al. The interconnection with inadequate amounts of available water, deteriorating water quality, failures in food security, and land degradation, as well as population exclusion due to climate, geography, soil type, latitude, and native vegetation [465], water deficits that would remain in water scarce regions aiming at food self- sufficiency, how those water deficits may be met by food imports and the cropland expansion required in low income countries without the needed purchasing power for such imports have been discussed in [484] and in [14,229]. More recent approaches are in Wada [485], the U.N. 2018 Water Report [486] and its assessment by Boretti et al. [487] and the 2019 U.N. Water Report [488].

As things stand and looking to the future, India and China, the main population components of BRICs, which have stable and organized regimes and economies, may move into the future somewhere between hybrid forms of SSP2 (moderate population growth)

and SSP1 (low population growth) but Africa and MENA, do not seem to be destined to continue on their present trajectories. Of great interest is the future of the ROW (Rest of the World) countries, which are those that do not belong either to OECD or BRICs, as in 1962 they had a population of 579.6 million which grew by 2010 to 1290 million at an average growth rate of 2.55% per year and have yet to be examined on a country-by-country basis to start drawing conclusions.

## 4. Discussion

### 4.1. Problems with Definitions

Water is an economic good [1,2] and so are its main components, blue and green water, and therefore, in any "supply–demand" definitions, as in the 2007 U.N. definition in Section 3.1.3, the economic interpretation is on equal footing with any physical one as water allocation, which may induce scarcity for agricultural use, is within the purview of central economic policy, e.g., via use-dependent price discrimination. All definitions should be explicitly limited to available water resources, as unavailability is usually accompanied by incurring the cost of making water available, in terms of money, time and the environmental impact of materials employed via their water footprint, which differentiates it from already available water. The supply–demand approach, as in [47,72], where imbalance happens at "prevailing prices" [79–81], is a bit nebulous as the automatic reaction of an open market in the case of stakeholders (Figure 5) is to seek equilibrium at a higher product price, traversing the stakeholder layers horizontally and/or in the market for alternative uses. In fact, if after a bad season due to water scarcity agriculture adopts a growth pattern to return to previous levels and is henceforth subject to uneven water scarcity, then prices will rise in Baumol's model [489].

In fact, the definition of water in the Dublin Water First Principle, "Water is a finite, vulnerable and essential resource which should be managed in an integrated manner" leads indirectly to that of a Global Public Good defined as "issues that are broadly conceived as important to the international community, that for the most part cannot or will not be adequately addressed by individual countries acting alone and that are defined through a broad international consensus or a legitimate process of decision-making" [490]. Yet, this broad international consensus is contrary to water rich sovereign country's interests and hence its materialization is highly improbable without counterbalancing conditions being imposed to importers.

As Winpenny points out "There are degrees of scarcity" among which "need" is included, the lower bound of the scarcity bandwidth, which has a dual interpretation, that of actual physical need, which represents the closing of the gap between "what is" and "what ought to be" from an objective point of view and that of "felt need", which is based on people's subjective opinions and perceived trends and outlooks [491]. More to the point is Beatty's definition of need "the measurable discrepancy existing between a present state of affairs and a desired state of affairs as asserted either by an "owner" of need ("motivational need") or an "authority" on need ("prescriptive need") [492] while, regarding consumers, Samuelson distinguishes between what people "really want and need" [493] (p. 4) and particular attention should be paid to Thaler's economic theory of the consumer [494] and his position on perception of consumer utility maximization [495]. However, 'need' for humans, the water end users, then requires per capita quantification to be defined, on a short-term basis, a medium-term basis and a long-term basis, where crops production is included, as seen below and in [496], as well as in the case of disasters [497] while Gleick et al [498] recommend 50 L/capita/day but include studies where the bare minimum for survival is 1.8–5.0 L/capita/day. As can be seen below in Figure 10 short-, medium- and long-term levels start from high quality drinking and cooking and end in decreased quality crops production livestock and recreational use in gardens.

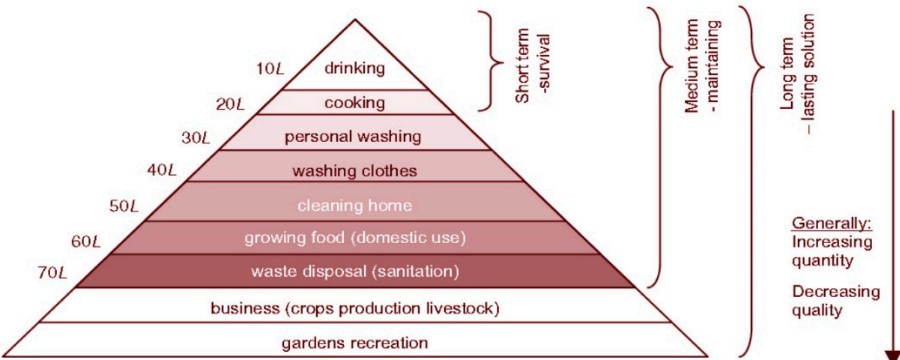

**Figure 10.** Short, medium and long term levels (modified from [499]).

More narrow types of water scarcity exist e.g., "managerial scarcity" due to inadequate management and maintenance of water resources leading to water scarcity [500,501], "institutional scarcity" caused by lack of institutional capacity to maintain and manage water resources [502], or institutional setting not flexible enough to accommodate changes [503], again leading to water scarcity and "political scarcity" where exclusion is politically imposed or economic policies that lead to scarcity are politically motivated [504].

Water use in terms of agriculture has peaks of demand during the growing season and is therefore a time dependent event which in turn forces demand and supply to be time dependent as well and not static or averaged as most of the scarcity definitions indirectly imply.

Blue water is not separated into water available for use (70%) and water reserved for supporting ecology sustainability (30%), which, if taken as separate entities, would face scarcity conditions separately without complementarity related balancing, but are treated as one entity leading to the assumption that they have the exact same impact, which they obviously do not. Moreover, there is no concept of a hierarchy of demand, set according to globally recognized dangers, e.g., by setting the present and upcoming challenge of food as a priority in terms of water scarcity.

The 2007 U.N. definition does not cover the case where water scarcity may occur in countries with the comparative advantage of low agricultural wages which are usually facing an export-import/balance-of-payments problem and increased food imports, most of which with high virtual water content, redeemable only by the increase of cultivated land and the corresponding increase in water consumption.

The existence of limits such as not using non-renewable water resources, or extending withdrawals beyond the renewable part of those that are renewable, and the percentage allocated to supporting ecology sustainability [505], while being physically correct are within the purview of the economic policy of sovereign states, which, according to their perception of need based primary objectives of their economic policy, may choose to directly or indirectly ignore them.

*4.2. Inequitable Availability*

Resource availability is important and there exist methods, e.g., entropy-based assessment [506] and indices [147], for its measurement, while in [507] blue water availability is defined as total natural runoff net of 20% assigned to environmental flow requirements and in [60] it is defined as total natural runoff plus groundwater, while in [55] the previous sum is considered net of 30% assigned to environmental flow requirements. Green water availability is defined in [55] as total rainfall infiltration in agricultural land minus runoff from this area multiplied by a reduction factor for minimum evaporation losses in agriculture of 0.85. Further, there is a large number of applicative papers on blue and green water resources availability as in Iran [508], in Africa [509], water resources availability is estimated at HRU, river catchment and city/region scales in the Wei River basin, China [510], in an Iranian data scarce watershed using SWAT [511], in the quantitative analysis of provision

probability [152], county-based green water availability for the entire U.S. [512], under climate change in the Beninese Basin of the Niger River Basin, West Africa [513], availability under climate change scenarios in the Mékrou Basin, Benin [514] and blue water estimates of current and future availability in Europe [515]. Availability indices for both blue and green water are listed in [516]. Groundwater availability is difficult to determine as there is bilateral flow connecting groundwater with blue water and the question of economic feasibility comes into play [517], e.g., there is groundwater in North Africa that is expensive to withdrawal from 150–200 m. boreholes, as pumps may be prohibitive for the prime user group of smallholders, and cost 100,000 USD plus upkeep costs [518]. A regional project may exceed 15 billion USD at 2012 purchasing power parity and levels [519] (while the 2019 national debt/GDP ratio for Tunisia is 72.33%, Morocco 65.77%, Algeria 46.28% and Libya is, in effect, a failed state), in the U.S. however, with a much healthier economy, the average well depth, where well water accounts for >50% of farm irrigation and all of it is pump extracted, is 72 m. [520]. Future projections for a 2 °C global climate lead to the conclusion that blue water will show increasingly uneven runoff distribution, which, unless water storage infrastructure is increased, will turn into floods [521].

*4.3. Inequitable Accessibility*

One of the current key problems is inequitable accessibility to water stocks and flows, as many flowing supply sources are sequestered from general dissemination for use due to their adverse location, in terms of population or/and productive land distribution and industry e.g., the Amazon, amounting to 16–20% of global runoff [522], has a 95% globally inaccessible flow. The local population to which the water is available does not exceed 35 mil. and Zaire-Congo has 50%, both being exoreic rivers [523]. The adverse localization of water resources extends to whole countries suffering from water deficit [524]; early studies did not include in-country variations [32], while it seems to be centred in the belt around 10° to 40° northern latitude [525] and, in addition, water availability is generally considered to be highly variable over space and time [526]. This uneven spatial distribution of water is manifest in the fact that South and Central Europe has 24–30% withdrawals, while in Northern Europe it may be less than 3% [53]. By defining arid zones (hyperarid, arid, semiarid and dry) as those where the rate of evaporation is greater than the rate of precipitation, which have a long geographical history [527], it is found that they cover 41% of world landmass with a third of the world's population [528], where 50% of the world's livestock is raised and 44% of the world's food is cultivated, exclusive of water demanding agricultural produce [529]. On the other hand, 60% of freshwater is located in nine countries, Brazil, Canada, China, Colombia, the Democratic Republic of Congo, India, Indonesia, Russia (which owns, along with Mongolia, Lake Baikal containing 20% of global unfrozen water [530,531]) and the U.S. [532], and these countries additively constitute 44% of world landmass with 50% of the world population [472]. Renewable resources are concentrated in Brazil (8233 km$^3$), Canada (2902 km$^3$), China (2840 km$^3$), Colombia (2132 km$^3$), India (1911 km$^3$), Indonesia (2019 km$^3$), Peru (1913 km$^3$), Venezuela (1320 km$^3$), Russia (4508 km$^3$) and the US (3069 km$^3$) [533]. Still, these countries have internal arid zones e.g., the Southwestern U.S. arid regions [534] and desert southwest [535]. Total inaccessibility of rivers, including the Amazon, the Zaire-Congo and remote rivers of North America and Eurasia which have no dams at all [536] lead to a totally inaccessible remote flow of 7774 km$^3$/year, amounting to 19% of total global annual runoff [537]. Inadequate accessibility can happen even in countries with plentiful available water as in the case of Nepal [538].

*4.4. Blue Water Loss*

If blue water discharges from exoreic rivers into the ocean, control of its re-entry into the land system is removed from the region it came from and passes on to the ocean water evaporation system, which may redistribute it, in part or in whole, to other regions. Thus, this may be considered to be a loss of blue water in the regional sense, only but not globally.

In terms of coastal regions without any exoreic river input to the sea, land precipitation as a function of distance drops at 300 km from the coast to 750 mm yr$^{-1}$ from 1300 mm yr$^{-1}$ at the coast line [539], in a recent study, the numbers differed, by taking 1931–2010 average, the decline is from a 50 km land coastal zone at 911.5 mm yr$^{-1}$ to 727.2 mm yr$^{-1}$ in the 100 to 150 km off-coast zone [540]. On the other hand, in the almost unique case of the Amazon river, a simulation with and without Amazon discharge into the Atlantic leads to the generation of an impact via teleconnections on the North American and European climates by inducing a NAO phase change [541]. This verifies the initial statement of this section but with an additional proviso. The 2010 Amazon drought was caused in part by increasing Pacific Sea surface temperatures (SSTs) of which the increase was triggered by the low salinity Amazon plume itself, created by the river's discharge into the ocean [542], which may intensify El Niño Southern Oscillation events and, most importantly, associated periodic Amazon droughts [543]. This last impact, in effect, describes a partially self-reaction process caused by the discharge to sea, which is detrimental to the river flow, itself enhancing the regional loss from the discharge. It should be noted that according to an initial estimate, globally 37,288 ± 662 km$^3$/year of water amounting to 35% of terrestrial rain are discharged from exorheic rivers into the sea [544] and in a latter one 45,500 km$^3$/year [545]. For the global ocean, minus the Arctic, this discharge shows small or downward trends for the largest 200 rivers [546] while a study of the 50 top rivers shows that 57.5% has a downward trend and 42.5% an upward trend, mainly due to climatic conditions [547]. Average annual discharge of freshwater from six of the largest Eurasian rivers to the Arctic Ocean has increased by 7% from 1936 to 1999 [548]. Additionally, uncaptured floodwater amounts to 20,426 km$^3$/year [537,549] and extreme precipitation leads to increased flood frequency [550]. Flood frequency is analysed by various, including model analysis [551] which are compared in [552], estimation methods were applied in Europe [553], uncertainties were examined in Norway [554] and in China [555], sensitivity for data record, statistical model, and parameter estimation methods in the U.S. [556] and at regional level for Mediterranean basins [557]. Climate change influences flood frequency [558] and is accounted for quantitatively in flood frequency analysis [559]. This is seen in, among others cases, Iran [560], the U.K. [561], the Yangtze basin, China [562] and in the case of extreme floods in Finland [563].

### 4.5. Unevenly Distributed Precipitation

In addition, annual precipitation is unevenly spatially distributed and across seasons leading to green water distortions, impacting agricultural production, as in smallholder and subsistence agriculture [564], where atmospheric warming leads to the amplification of precipitation extremes [198,565], as seen in trends in research on global climate change [566], in changing climate inducing time-varying extreme rainfall intensity-duration-frequency curves [567]. Also in China as Minjiang River annual and seasonal precipitation variation [568], in annual and daily extreme precipitation distribution trends over 1960–2010 in urban areas [569], in changes in precipitation patterns over Beijing over 1960–2012 (which is a 21.54 million 'thermal island') [570], annual/seasonal precipitation variation in a mountain area [571], in Serbia as annual and seasonal variability of precipitation in Vojvodina [572], 1961–1990 mapping of annual precipitation [573], in the impact of climatic factors on maize yields [574]. Further, in Iran as in 1950–2000 rainfall trends [575], using 1950–2000 spatial and temporal variability of precipitation [576], annual and seasonal distribution pattern of rainfall including neighbouring regions [577], in India as in trends in the rainfall pattern over the whole country [578], uneven distributions overturn benefits of higher precipitation for crop yields [579], changes in rainfall seasonality pattern over the whole country [580], in Africa in 19th through the 21st century rainfall over the continent [581], and in Mexico as in disruptions in the timing and intensity of precipitation in Calakmul [582]. This will extend to highly populated centres in one form or another, e.g., in the U.S. such as in quantification of changes in future intensity-duration-frequency curves [583]. Uneven precipitation, such as the one projected in India, in terms of fluctua-

tions in water flows and changing monsoon patterns, leading to blue water waste due to low levels of water storage capacity per capita [584].

### 4.6. Climate Uncertainty

Climate is defined in terms of the physics of its change and variability as "a forced, dissipative, chaotic system that is out of equilibrium and whose complex natural variability arises from the interplay of positive and negative feedbacks, instabilities and saturation mechanisms" [585]. Climate change and variability uncertainty is a subject which has not been attributed sufficient importance [586] in applicative studies and this uncertainty is directly related to existing [587] hydrological uncertainty [588], such as in impacts on the hydrological cycle over France with uncertainties taken into consideration [589] and has a direct impact on water resources [590]. Those uncertainties are associated with internal climate system variability and the hydrologic modelling itself [591], while climate forcing uncertainty impacts on the variations of global and continental water balance components [592] extending even into environmental flows e.g., for the Mekong River [593]. They are of great importance as the existence and size of uncertainty, distinguished into uncertainty due to lack of knowledge regarding the changes and uncertainty due to variability [594], which is critically associated with risk assessments connected to instrumental economic variables to which physical uncertainties must be transmitted in some way and therefore their detailed and credible knowledge will enable at first instance to imbed some flexibility in water infrastructure programs [595]. One opinion is that this transmission can be manifested by changing decision rules and evaluation principles used for water infrastructure project justification so that they are compatible with climate uncertainty [596]. However, climate uncertainly, along with all other main variables, is incorporated into wider scenarios with distinctly different impacts, which are deterministically manifested during lengthy periods, a fact that more or less precludes decision-making regarding proactive long-term projects. Perhaps a promising approach is dynamic adaptive policy pathways (DAPP) [597].

There are methods leading to the separation of deterministic and probabilistic components of main variables [598] e.g., in the case of precipitation [599], in hydrologic time series [600] and in general e.g., in the case of time series [601,602] but, in separate models, they do not match the success of a hybrid model [602]. It should be noted that these methods separate uncertainty from determinism, but they do not cause any reduction to it.

Added to the above is the case of for climate applicable [586,603–605] Knightian uncertainties, "true uncertainties" indetectable under hindsight which are not reducible to an "objective, quantitatively determined probability" [606] (p. 321). These are manifestly transmitted through to any economic assessment process, in addition to entropic preferences [607], leading to the adoption of one-parameter Atkinson preference functions [608] which may be problematic to the classic style Georgescu–Roegen entropy, linking climate to economy [609].

### 4.7. Country Scale More Pertinent Than Global

At a global scale, resource economics is considered to be a thermodynamically closed system where entropy increases or energy degrades [609,610] and so is the hydrological cycle [611,612]. As seen in Section 3.5.2 3990 billion cubic meters of water were consumed globally in 2014, a quantity which approaches the lower limit of the 4000–6000 billion cubic m/year danger zone according to the planetary boundaries framework [613]. However, this picture, taking into consideration that world population in 2014 was 7.2 billion [614], yields an average per capita consumption correspondence which would have meaning if population distribution and the subjects addressed in Section 4.2, Section 4.3, Section 4.4, Section 4.5 were equitably distributed, which is not the case.

Renewable freshwater resources per capita differ according to regions as can be seen in Figure 11.

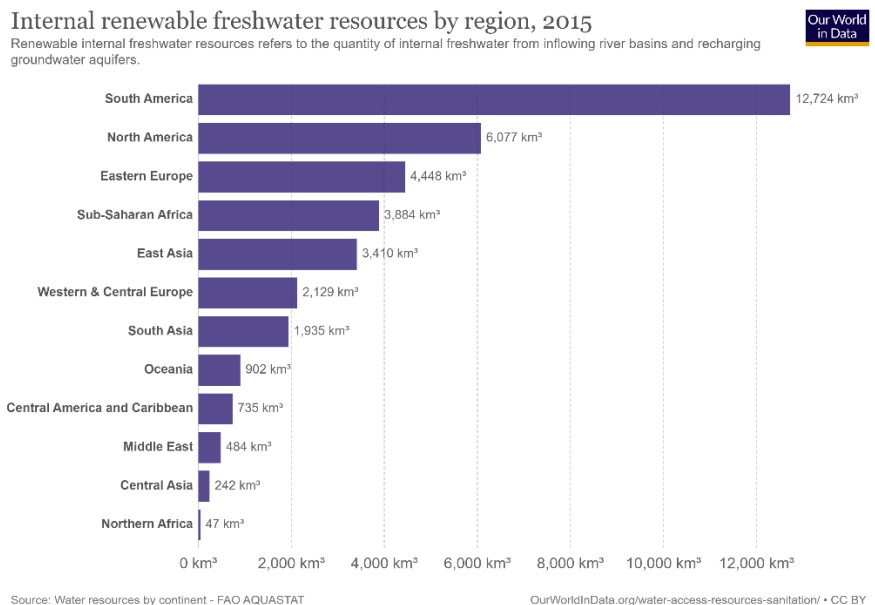

**Figure 11.** Renewable freshwater per region 2015 (modified from [615]).

On the other hand, both resource economics, dictated by the sovereign state's economic policy, and the hydrological cycle are open systems at country level and consequently the degrees of water scarcity differ as well. In this case, as can be seen at country level in Figure 19 below, the notion of relative scarcity pioneered by Faber [616], defined in terms of quantity over available water resources as "a good is scarce in relation to other scarce goods" [617], plays a role which cannot manifest itself in a credible way at a global scale due to inequitable availability of resources. In the case of transnationally shared resources a case study of the Aral Sea [618] shows equal probability of conflict and cooperation, a matter not taken into account at global scale statistics and as a general problem it is still in the phase of suggesting a viable solution [619–621].

The summary legal construct of international treaties and U.N. decisions regarding the prevention of water scarcity have no "teeth" as transgressor countries are not penalized in any meaningful way while supervision has only a fact-finding mandate. In reality there is no global organization with the power to enforce any of the above and assume responsibility for the results of these decisions which in essence means that in the applicative phase in the immortal words of U.K. Supreme Court Judge Thurlow, " . . . has no soul to be damned and no body to be kicked" just like a corporation [622] hold true.

Increased population and low GDP lead to a tipping point, regardless of perceived estimation water resources being ample [623] while conversely the water footprint expands with higher GDP [624]. The point of view where microscale precedes mesoscale is also supported by Falkenmark [77]. A list of country rankings regarding water risk is found at Aqueduct [625].

In Figure 12 it is shown that most countries belonging to the set with available data are in states ranging from low-to-medium stress to extremely high stress.

At the country level, in the common case of a water market subject to "institutional arrangements", we enter into the realm of state economic policy where restrictions are imposed while the central government budget might cover, at least in part, the reallocation cost as the economy will suffer from the trade deficit incurred by virtual water imports, as seen in an 160 country study over the period 1982–2007, where drought events increase net global virtual water flows by $5 \times 10^9$ m$^3$ yr$^{-1}$ to $6.34 \times 10^9$ m$^3$ yr$^{-1}$ while each additional square kilometer of agricultural land area reduces net virtual water import by 10,620 to 18,419 m$^3$ [626], a 2010 estimate leads to that international trade reduces global water use in agriculture by 5% [627], while in some Mediterranean countries, a reduction of 1% of agricultural productivity corresponds to imports of 233 million cubic meters of virtual

water [628]. However, virtual water import/export balance may not correspond to water scarcity only, case in point being the grain import-export balance in Spain in the 1997–2005 period [629], a country whose south-eastern part is semi-arid [630] and water scarcity appears every 5–6 years [631]. Quite importantly, virtual water introduces a hitherto unknown and hence Knightian type variable [606], the "virtual water cycle" variable [632]. All these quantifiable additions on instrumental economic variables would pass unseen at a position over country level.

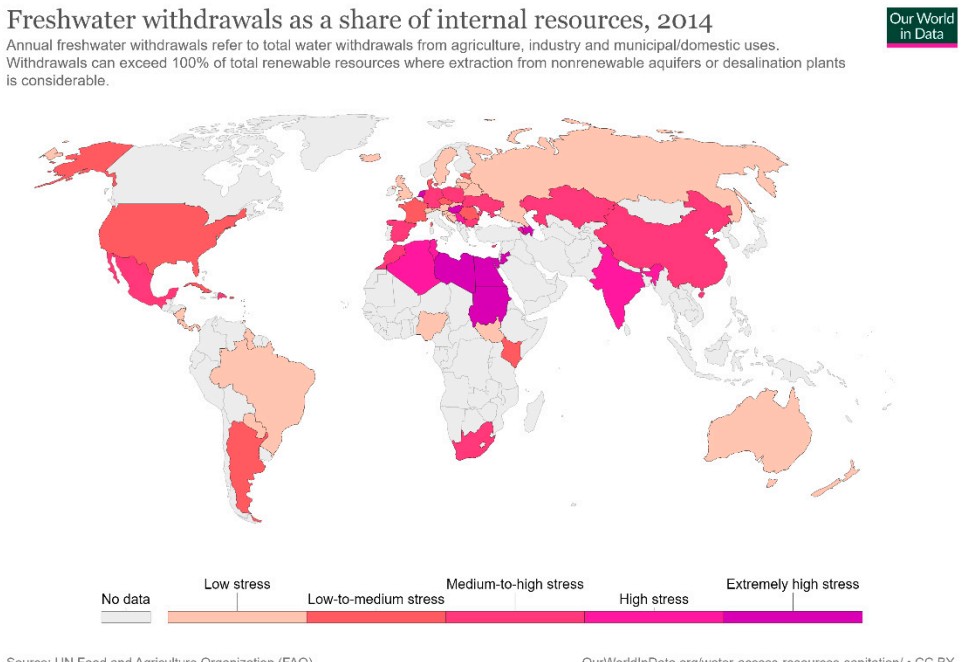

**Figure 12.** Freshwater withdrawals as a share of internal resources (modified from [615]).

Thus, the country-by-country depiction which shows anisotropy allows for direct economic assessment in the spirit described in the Methodology section while the global statistical picture does not.

## 5. Conclusions

The objective of this paper was analysed in seven stages, using a substantial number of references which enabled the decomposition of general and theoretical definitions and physical variables into their economically important facets, which were firmly anchored temporally, spatially, and in terms of scale, through the use of a large number of referenced incidents. Particular care was demonstrated in focusing on present and future problems stemming either from external conditions or from the internal structures of the theories which must be carried over to a faithful economic assessment of the scarcity phenomenon.

Difficulties emerge in the matter of associating physical blue and green water scarcity to the instrumental variables causing it. The first is the set of definitions describing the physical phenomenon of scarcity which was analysed in this paper. The definitions are too broad to be meaningfully carried over to a realistic economic model and the lack of distinction between "want" and "need" causes problems of economic interpretation. The second one is that these variables are interdependent to some degree via the physical theory of which they are part without a clear formal and theoretical quantitative and qualitive analysis of this interdependence, which causes problems in the economic depiction of scarcity in a process of economic reverse engineering seeking causal connection i.e., the case where one seeks to depict in economic terms any form of relational correspondence between a set of scarcity causing variables in the process of triggering scarcity and scarcity itself. This difficulty is quite important as it presents impediments to any economic theory

from going down to the basic platform underlying this type of scarcity and linking global variables to scaled down instances in a seamless unified form as scaling down introduces new variables which operate at the particular level of scale and have no global existence, which perhaps has a solution by assuming the existence of a set of variables which at global level may appear as noise or Knightian variables but as downscaling occurs they acquire a concrete form allowing for a hybrid deterministic/uncertainty depiction. The third is the fact that country level climate uncertainty or causality has no clear picture which impedes correspondence between physical and economic uncertainly, the latter leading to economic risk, which was analysed extensively in this paper.

From the physical point of view the emerging picture is bleak but then the incident instances reported and the interrelationship of the main variables are commensurate with this point of view, and if one digs deeper there is a chain of too many beneficial constraints assumed to be viable with a high degree of certainty and as many uncertainties not taken into consideration.

However, by assuming the country level water scarcity to be the dominant building block more targeted measurements at country level leading to a structured component picture at regional level will allow for statistical analysis, including maximum application of causality tests, which will clarify the empirical interdependence of physical scarcity causing variables and allow for establishing a clear-cut correspondence with economic instrumental variables.

### 6. Future Work

The next steps are the examination of economic scarcity in the way adopted in this paper and an economic model which ties both physical and economic scarcity with the main instrumental variables of a national economy.

**Author Contributions:** Conceptualization, K.Z. and D.P.; methodology, K.Z. and D.P.; validation, K.Z. and D.P.; formal analysis, K.Z. and D.P.; investigation, K.Z. and D.P.; resources, K.Z. and D.P.; data curation, K.Z. and D.P.; writing—original draft preparation, K.Z. and D.P.; writing—review and editing, K.Z. and D.P.; visualization, K.Z. and D.P.; supervision, D.P.; All authors have read and agreed to the published version of the manuscript.

**Funding:** This research received no external funding.

**Institutional Review Board Statement:** Not applicable.

**Informed Consent Statement:** Not applicable.

**Data Availability Statement:** Data sharing not applicable.

**Conflicts of Interest:** The authors declare no conflict of interest.

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
