# Peer review of "An In-Depth Analysis of Physical Blue and Green Water Scarcity in Agriculture in Terms of Causes and Events and Perceived Amenability to Economic Interpretation"

_water, doi:10.3390/w13121693_

Round 1

Reviewer 1 Report

The authors have attempted to address and re-work the manuscript, and there are some good improvements made (e.g. parts of the discussion, removing some figures), but the paper is still relatively difficult to follow, and very long, with language issues and lack of flow. I encourage the authors to send the paper for language/flow editing, which could substantially help improve the readability. I've given some suggestions below, which at minimum, needs to be addressed before I think this could be at the level for publication:

  • The second paragraph (sentence starting with: “Water scarcity is considered to be the result of a complex interaction of social…”) of the introduction reads very strange, and this one sentence is almost the full paragraph – please shorten and clarify.
  • Unfortunately, the language (both spelling and flow) need improvement – as of now, the lack of proper sentence structure makes it difficult to read the paper. Maybe it would be worth sending it to language editorial service?
  • The objective of this paper is to present Blue and Green water scarcity by general case – what does this mean? “general case”? Can you better explain this in the objective?
  • There is no proper reference in the text to Fig. 3, Fig. 4… etc (please check and update this throughout). and the figure legends are not sufficient – if all these figures are needed (which I doubt), they need to be properly introduced and explained – otherwise its better to remove them.
  • Section 3.2.1 why is water reserves compared to mineral reserves? I really don’t see the value of this comparison. Actually, I think the paper is better without this section at all, as most of it is basic knowledge and not needed for the paper (e.g. water partitioning on earth). I would actually also recommend to remove section 3.2.2. as well, and put 3.3 after 3.1.3 as that builds more on the flow of the water scarcity focus

Author Response

Thank you for your most constructive comments. Partial use only of the “Track Changes” function is possible as the Referencing Program used in not that of MS Word and Track Changes involving the IEEE notation will not automatically renumber the sources quoted in the text or delete those no longer necessary in the Reference section.

  1. The second paragraph has been reformed so as to enhance and clarify the meaning it is intended to convey.
  2. Spelling has been checked and corrected but the language use is not faulty as it is merely the way we write bearing in mind that the article is intended primarily for economists. Please check [1] as an example of previously published work.
  3. The ‘general case’ conundrum has been resolved by the addition of the previously omitted word ‘both’
  4. Some figures were deleted and the remaining ones are addressed as per your suggestion.
  5. Section 3.2 has been erased as per your suggestion
  6. As a matter of record only, as the section containing it has been erased, water reserves are comparable to mineral reserves. According to [2] soil moisture is assessed to be 16.5 x 103 km3 and according to [3] fossil aquifer water reserves exceed 92 x 103km3 (5.57 times soil moisture). That makes fossil aquifer water reserves significant in terms of size and in [3] (Appendix A) Groundwater mining technology and cost function are analyzed. A more recent study [4] augments the significance of fossil aquifer water reserves estimating that fossil groundwaters comprise a large share (42–85%) of total aquifer storage in the upper 1 km of the crust, and the majority of waters pumped from wells deeper than 250 m. There is a common denominator between fossil aquifer water reserves and mineral reserves : extraction diminishes the source content without the prospect of in-source replacement.

References

[1]        K. Zisopoulou et al., “Recasting of the WEF Nexus as an actor with a new economic platform and management model,” Energy Policy, vol. 119, pp. 123–139, 2018.

[2]        I. A. Shiklomanov, “World fresh water resources.,” in Water in Crisis: A Guide to the World’s Fresh Water Resources, P. H. Gleick, Ed. New York: Oxford University Press, 1993, pp. 13–24.

[3]        Y. Tsur, H. Park, and A. Issar, “Fossil groundwater resources as a basis for arid zone development? An economic inquiry,” Int. J. Water Resour. Dev., vol. 5, no. 3, pp. 191–201, 1989.

[4]        S. Jasechko et al., “Global aquifers dominated by fossil groundwaters but wells vulnerable to modern contamination,” Nat. Geosci., vol. 10, no. 6, pp. 425–429, 2017.

Reviewer 2 Report

The paper has been structurally improved according to the reviewers' comments. Now, the current version of the manuscript can be accepted for publication.

Author Response

Thank you for your most supporting comments. 

Round 2

Reviewer 1 Report

Thank you for addressing some of my concerns. I think the paper is of sufficiency for publication in Water.